

**Inter-comparison of online and offline methods for measuring ambient heavy and trace elements**
**and water-soluble inorganic ions (NO3-, SO42-, NH4+ and Cl-) in PM2.5 over a heavily polluted**
**megacity, Delhi**
Himadri Sekhar Bhowmik[1], Ashutosh Shukla[1], Vipul Lalchandani[1], Jay Dave[3], Neeraj Rastogi[3], Mayank
Kumar[4], Vikram Singh[5], and Sachchida Nand Tripathi[2]
[1]Department of Civil Engineering, Indian Institute of Technology Kanpur, Kanpur, India
[2]Department of Civil Engineering and Centre for Environmental Science and Engineering, Indian Institute
of Technology Kanpur, Kanpur, India
[3]Geosciences Division, Physical Research Laboratory, Ahmedabad, India
[4]Department of Mechanical Engineering, Indian Institute of Technology Delhi, New Delhi, India
[5]Department of Chemical Engineering, Indian Institute of Technology Delhi, New Delhi, India
*Correspondence to:* S. N. Tripathi (snt@iitk.ac.in)
Keywords: Aerosol mass spectrometer (AMS), Xact 625i ambient metal mass monitor, Ion
Chromatography (IC), ICP-MS, elemental composition.
**Abstract**
Characterizing the chemical composition of ambient particulate matter (PM) provides valuable information
on the concentration of secondary species, toxic metals and assists in the validation of abatement
techniques. The chemical components of PM can be measured by sampling on filters and analysing them
in the laboratory or using real-time measurements of the species. It is important for the accuracy of the PM
monitoring networks that measurements from the offline and online methods are comparable and biases are
known. The concentrations of water-soluble inorganic ions (NO3-, SO42-, NH4+ and Cl-) in PM2.5 measured
from the 24 hrs filter samples using ion chromatography (IC) were compared with the online measurements
of inorganics from aerosol mass spectrometer (AMS) with a frequency of 2 mins. Also, the concentrations
of heavy and trace elements determined from the 24 hrs filter samples using inductively coupled plasma
mass spectroscopy (ICP-MS) were compared with the online measurements of half-hourly heavy and trace
metal's concentrations from Xact 625i ambient metal mass monitor.  The comparison was performed over
two seasons (summer and winter) characterized by their different metrological conditions at IITD and
during winter at IITMD, two sites located in Delhi, NCR, India, one of the heavily polluted urban areas in





the world. Collocated deployments of the instruments helped to quantify the differences between online
and offline measurements and evaluate the possible reasons for positive and negative biases. The slopes for
$SO_4^{2-}$ and $NH_4^+$ were closer to 1:1 line during winter and decreased during summer at both sites. The higher
concentrations on the filters were due to the formation of particulate $(NH_4)_2SO_4$. Filter-based $NO_3^-$
measurements were lower than online $NO_3^-$ during summer at IITD and winter at IITMD due to the volatile
nature of $NO_3^-$ from the filter substrate. Offline measured $Cl^-$ was consistently higher than AMS derived $Cl^-$
during summer and winter at both sites. Based on their comparability characteristics, elements were
grouped under 3 categories. The online element data were highly correlated ($R^2$>0.8) with the offline
measurements for Al, K, Ca, Ti, Zn, Mn, Fe, Ba, and Pb during summer at IITD and winter at both the sites.
The higher correlation coefficient demonstrated the precision of the measurements of these
elements by both Xact 625i and ICP-MS. Some of these elements showed higher Xact 625i elemental
concentrations than ICP-MS measurements by an average of 10-40% depending on the season and site. The
reasons for the differences in the concentration of the elements could be the distance between two inlets for
the two methods, line interference between two elements in Xact measurements, sampling strategy, variable
concentrations of elements in blank filters and digestion protocol for ICP measurements.

## 1. Introduction

The adverse effect of ambient particulate matter (PM) on human health and the role of PM in visibility
degradation, altering earth's radiation balance, and climate change has received global attention in the last
two decades (Pope et al., 2009; Hong et al., 2019; Wang et al., 2019). To gain better insight into their
properties, the chemical characterization of particulate matter and its source attribution is crucial. The
National Capital Region (NCR), which includes India's capital (New Delhi) along with some districts
(Gurugram, Faridabad, and Noida) of the adjoined states of Haryana, Rajasthan, and Uttar Pradesh, is one
of the most polluted urban areas in northern India with a population over 47 million (Bhowmik et al., 2021).
According to World Economic Forum, New Delhi was listed as the most polluted city globally, with an
annual average $PM_{2.5}$ concentration of ~140 μg m$^{-3}$ (World Health Organisation, 2018).  NCR has been a
specific area for researchers for the past couple of years due to its unprecedented $PM_{2.5}$ levels. Various large
and small scale industries, power plants, construction activities, and rapid increase in the vehicle numbers
(11 million in 2018) (Rai et al., 2020) are among the several causes that massively reduces the air quality
index (AQI) (Rai et al., 2020; Sharma & Kulshrestha, 2014). Further, the crop residue burning during the
month of Oct-Nov in the adjoining states of Haryana and Punjab on a larger scale worsens the air quality.
For decades, the mass concentrations of major water soluble inorganic ions (WSIS) and heavy and trace
metals in PM have been carried out by sampling them on filters and subsequently analysing them in



laboratory. Water-soluble inorganic ions (WSIS) and heavy and trace elements from these filter samples
are analyzed using ion chromatography (IC) (Bhowmik et al., 2021; Rengarajan et al., 2007; Rastogi &
Sarin, 2005) and inductively coupled plasma-mass spectroscopy (ICP-MS) or inductively coupled plasma-
optical emission spectroscopy (ICP-OES) (Patel et al., 2021) respectively. Usually, these filters are
collected over 24 hrs interval. Traditional receptor models usually use these offline measured data of very
low temporal resolution, making it challenging to characterize the short pollution episodes and dynamics
of pollution sources. Further, un-denuded filter sampling can have both negative and positive artifacts due
to volatile species (Lipfert, 1994; Pathak & Chan, 2005). The absorption of acidic and alkaline gases on the
filter substrates, if not removed prior to sampling, can give positive artifacts and result in overestimating
species concentration. Likewise, the evaporation of semi-volatile compounds (ammonium nitrate) from
filter substrates can give negative biases and result in underestimating aerosol concentration and its species
(Pathak & Chan, 2005; X. Zhang & Mcmurryt, 1992). The degree of artifacts can be affected by several
factors, including temperature, relative humidity, type of filter substrate, the aerosol loading on the filter
substrate, etc. Transient events can also lead to mismatch. To overcome the limitations of low temporal
resolution and avoid the artifacts associated with offline filter sampling, methods have been developed for
measuring aerosol chemical composition at a higher time resolution in the order of hours or minutes.
Aerosol Mass Spectrometer (AMS) (Canagratna et al., 2007; Jayne et al., 2000; Jimenez et al., 2003) is one
kind of such instrument, which provides size-resolved chemical composition of non-refractory submicron
aerosols, e.g., organics, sulphate, nitrate, ammonium, and chloride at the order of hours or even minutes.
For other important components, such as calcium (main constituents of soil dust and construction activities)
and potassium (tracer of biomass burning), which AMS cannot measure, Xact ambient metal mass monitor
can be used. It is capable of measuring 40 elements, e.g., Al, Si, P, S, Cl, K, Ca, Ti, V, Cr, Mn, Fe, Co, Ni,
Cu, Zn, Ga, Ge, As, Se, Br, Rb, Sr, Y, Zr, Nb, Mo, Pd, Ag, Cd, In, Sn, Sb, Te, I, Cs, Ba, La, Ce, Pt, Au,
Hg, Tl, Pb and Bi with a frequency of every 30 mins to 4 hours. However, the high time-resolution
instruments measure lower range of species concentrations with higher limit of detection (LOD) than the
offline based methods (Tremper et al., 2018).  Both offline and online methods have their own strength and
weaknesses. Uncertainties in offline filter analysis methods have been extensively studied (Pathak & Chan,
2005; Viana et al., 2006), but the novel online methods pose new problems (Wu & Wang, 2007). For
example, when the ambient concentrations are very low, online measurements are often close to the MDL
values due to the short integration times (Malaguti et al., 2015).
Previous studies in Delhi-NCR have used low time resolution filter-based methods for chemical
characterization of submicron aerosols (Bhowmik et al., 2020; Nagar et al., 2017; Pant et al., 2015; S. K.
Sharma et al., 2016; Singhai et al., 2017). On the other hand, there are only a few studies in Delhi-NCR



that used high time resolution methods (HR-ToF-AMS, Q-ACSM, Xact) for characterization and source
apportionment of coarse and fine particulate matters (Gani et al., 2019; Lalchandani et al., 2021; Rai et al.,
2020, 2021; Singh et al., 2021; Tobler et al., 2020). Online and offline measurements both have their
advantages and limitations. For both these measurements, the quality of the data highly depends on the
calibration of the instruments. For Xact, the multi-element mix standard might not represent ambient
elemental mix if the ambient particulate matters are too low or too high in concentration, affecting the
collection properties of the filter (Indresand et al., 2013). For filter-based water-soluble inorganic ion and
metal analysis, confidence in the data depends on the calibration as well as the volatility, solubility, and
digestion protocol used for the extraction of water-soluble inorganic ions and elements, respectively. Thus,
it is vital for monitoring networks that both the offline and online measurement methods give comparable
results. Few published studies have compared inorganics and elements from filter-based measurements and
semi-continuous methods (e.g., Furger et al., 2017; Nie et al., 2010). The inter-comparison in these studies
is unjustified for highly polluted areas, as the specie values observed in these studies are much below MDL
due to very low ambient concentration of secondary species and elements. It will be interesting to study the
inter-comparison in highly polluted areas. To the best of our knowledge, there are neither any published
seasonal and temporal comparisons of inorganics from high time resolution AMS measurements and filter-
based measurements from ion chromatography nor any comparisons of heavy and trace metals from high
time resolution Xact 625 ambient metal mass monitor and offline measurements from ICP-MS in the
heavily polluted Delhi NCR region.
This study demonstrates a comparison between online and offline measurements of WSIS and heavy and
trace metals at two sites in Delhi NCR during summer (June-July 2019), characterized by moderate levels
of local pollution and winter (October-December 2019), affected by high levels of pollution from local
sources and regional transport of crop residue burning emissions from adjoining state of Haryana and
Punjab.
**2. Methodology**
**2.1. Sampling sites**
Delhi-NCR, a highly polluted urban area with an annual average $PM_{2.5}$ concentration of 140 µg m$^{-3}$ and a
population of over 47 million, is surrounded by the Thar desert on its west and Indo-Gangetic Plain on its
east to south-east. The temperature is about ~35°C-48°C during summer (April-June), and winter
(December-February) is relatively cooler, with temperature ranges from ~2°C-15°C (Bhowmik et al., 2020;



Lalchandani et al., 2021; Tobler et al., 2020). The wind is mostly north-westerly during both summer and
winter.

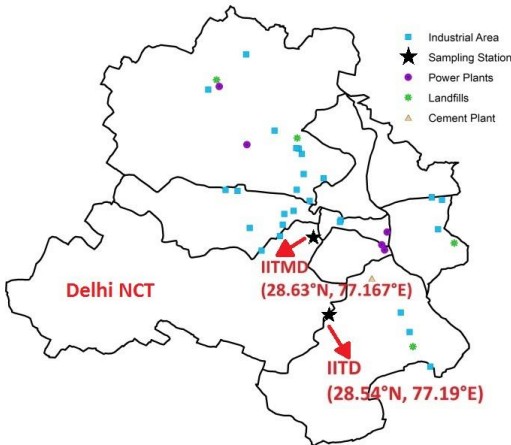


Fig.1. Sampling sites with various emission sources like power plants, industries, landfills, etc.

### 2.1.1. Delhi NCR site 1: IITD

High volume PM$_{2.5}$ samples were collected on the rooftop of the Centre for Atmospheric Science (CAS)
building at the Indian Institute of Technology, Delhi (IITD) (28.54°N, 77.19°E; ~218 m amsl) about ~15
m above the ground level. Further, one HR-ToF-AMS and Xact ambient metal mass monitor were deployed
inside a temperature-controlled laboratory on the 3$^{rd}$ floor of the same building about ~10 m above the
ground level. It is an educational institute cum residential campus having restaurants nearby and very close
to (<200 m) heavy traffic road. Lalchandani et al., (2021); Rai et al., (2020) observed source signatures of
emissions from industries, power plants, vehicles, and waste burning in this site.

### 2.1.2. Delhi NCR site 2: IITMD.

Offline PM$_{2.5}$ sampling was carried out on the rooftop of the main building at the Indian Institute of Tropical
Meteorology Delhi (IITMD) (28.63°N, 77.167°E; ~220 m above msl) about 15 m above the ground level.
Moreover, HR-ToF-AMS and Xact ambient metal mass monitor were installed inside a temperature-
controlled laboratory on the 2$^{nd}$ floor of the same building at the height of ~8 m from the ground level. This
site is placed in the central urban area of Delhi and surrounded by Central Ridge reserve forest and
residential areas (Tobler et al., 2020) and around 14 km away in North West direction from IITD.  A recent



study by Lalchandani et al. (2021) observed the site is dominated by emissions from traffic, solid fuel
burning, and oxidized organic aerosols. The location of the sampling sites are shown in Fig. 1.
**2.2. Sampling details**
**2.2.1 Offline sampling**
Biweekly 24 hrs (January- September 2019) and daily 24 h (October- December 2019) $PM_{2.5}$ samples were
collected on quartz filter substrates (Whatman; $8 \times 12$ inches) using a high-volume sampler (HVS) with a
flow rate of 1.13 $m^3$/min. Blanks were collected in the field by placing a fresh filter in the sampler while
not running. The collected filters, including field blanks, were zip-locked and stored in the freezer at each
site and periodically transported to CESE (Centre for Environmental Science and Engineering), IIT Kanpur,
where they were further stored at -20˚C in a deep freezer prior to analysis. For this study, the samples  from
the common period of offline and online sampling (October-December 2019) from the two sites were
analyzed for WSIS ($SO_4^{2-}$, $NO_3^-$, $NH_4^+$ and $Cl^-$) using an IC, and 32 metals (Al, Na, K, Ca, Ti, V, Cr, Mn,
Fe, Ni, Cu, Zn, As, Se, Rb, Sr, Zr, Cd, Sn, Sb, Ba, Pb, Cs, La, Ce, Pt, Tl, Mg, Li, Mo, Co and Pd) using an
ICP-MS. More details of analytical procedures are given in the instrumentation section.
**2.2.2 Online sampling**
At both IITD and IITMD site, high-resolution time-of-flight aerosol mass spectrometers (HR-ToF-AMS,
Aerodyne Research Inc., Billerica, MA) (Canagaratna et al., 2007; DeCarlo et al., 2006), equipped with
$PM_{2.5}$ aerodynamic lens (Peck et al., 2016) (Aerodyne Research Inc., Billerica, MA, USA) were installed.
Ambient fine particulate matters were sampled through $PM_{2.5}$ cyclone (BGI, Mesa Labs. Inc.) inlet at IITD
and through black silicon tubing at IITMD, placed ~1.5-2 m above the rooftop with a flow rate of 5 l/min
using long stainless-steel tubing (12 mm O.D). An aerosol sample dryer system (Aerodyne research, Inc)
was used to dry the ambient aerosols to maintain the output RH at 20%. At IITD, data were collected from
12th October 2019-31st December 2019 and 2nd June 2019-21st July 2019 during winter and summer
campaigns respectively. The data between 1st November and 14th November was not available, due to
hardware issues in the AMS during that period. At IITMD, data were only collected during winter campaign
from 25th October 2019 to 31st December 2019.
Two Xact 625i ambient metal monitors (Cooper Environmental Services, Beaverton, Oregon, USA) were
installed at IITD and IITMD. Ambient aerosols were sampled through a $PM_{2.5}$ inlet with a flow rate of 16.7
lpm. At IITD, sampling was carried out from 1st October 2019-31st December 2019 and 30th May  2019-25th
July 2019 during the winter and summer campaign, respectively. However, data between 16th July and 24th


July were not available due to hardware breakdown. At IITMD, samples were collected from 1$^{st}$ October
2019-31$^{st}$ December  2019 but data from 18$^{th}$ November to 26$^{th}$ November 2019 and 30$^{th}$ November to 14$^{th}$
December 2019 were not available due to instrumental problems.
Online measurements of inorganic ions ($SO_4^{2-}$, $NO_3^-$, $NH_4^+$ and $Cl^-$) from HR-ToF-AMS were compared
with the WSIS using an IC. Parallelly, heavy and trace metals obtained from Xact ambient metal mass
monitor were compared with the metal data from the offline filter measurements using an ICP-MS. Though
the sampling periods of AMS, Xact, and HVS were different for different seasons and different sites as well
(Table 1), only the common periods of online and offline sampling have been discussed in this study for
comparison.
Table 1. Sampling strategy and instrumentation used.

| | Interval | IITD | IITMD |
|---|---|---|---|
| **Quartz filter Sampling** | 3 days | January-September 2019 | January-September 2019 |
| | 24 hrs | October-December 2019 | October-December 2019 |
| **HR-Tof-AMS** | 2 mins | 2$^{nd}$ June-21$^{st}$ July 2019 | 25$^{th}$ October-31$^{st}$ December 2019 |
| | | 12$^{nd}$ October-31$^{st}$ December 2019 | |
| **Xact** | 30 mins | 30$^{th}$ May-25$^{th}$ July 2019 | 1$^{st}$ October-31$^{st}$ December 2019 |
| | | 1$^{st}$ October-31$^{st}$ December 2019 | |

Filters from the common periods were analysed for WSIS and heavy and trace metals using an IC and ICP-MS respectively.
**2.3. Instrument details**
**2.3.1 WSIS measurements by IC and HR-ToF-AMS**
For WSIS analysis, 9 sq.cm punch area of each collected filter was soaked in 30 ml of high purity milli-q
water (resistivity-18.20 MΩ cm) for 12 hours in pre-cleaned borosilicate test tubes to ensure maximum
solubility. The amount of water added, soaking time, etc. effect the solubility of the ions as well as the
extent of extraction. Details can be found in our previous paper (Bhowmik et al., 2020). Soaked samples
were filtered through 0.22 µm quartz filter papers to remove any suspended contaminations after an
ultrasonication for 50 mins. $Cl^-$, $NO_3^-$, $SO_4^{2-}$ and, $NH_4^+$ were measured for all the filter extracts by an IC
(Metrohm 883 Basic IC plus for cations and 882 compact IC plus for anions). Separate columns for the
analysis of cations and anions were installed in two separate modules. For anion and cation separation, an



AS 5-250/4.0 chromatography column and C-6 column was used, respectively. The sample carried by 3.2
mM $Na_2CO_3$+1 mM $NaHCO_3$ solution and 2.7 mM $HNO_3$ solution in the anion and cation module
separately passes through the charged columns to analyze each ion according to their polarity. The
calibration was performed by a seven-point method with a range of standards prepared by the serial dilution
from the stock solution standard of 10 ppm purchased from Metrohm. The uncertainty of the water-soluble
inorganic ions measured by IC was estimated as 4% (coverage factor~2) by the approach described in
Yardley et al. (2007).
HR-ToF-AMS measures size-resolved mass spectra of non-refractory particles (PM components that
vaporize at 600°C and $10^{-5}$ Torr, e.g., organics, nitrate, sulphate, ammonium, and some chlorides) of
submicron particulate matters. The details of this instrument can be found elsewhere (DeCarlo et al., 2006).
Briefly, ambient aerosols are collected through an orifice of 100 μm diameter and focused into a narrow
particle beam by an aerodynamic lens system installed inside the instrument, which has a transmission
efficiency of > 50% for $PM_{2.5}$ (DeCarlo et al., 2006; Peck et al., 2016). The particle size is determined after
analyzing the time of flight, i.e., time taken to travel along the length of the sizing chamber. The NR-PMs
are then vaporized by hitting the vaporizer at 600°C and at a vacuum of $10^{-7}$ Torr. Further, the vaporized
molecules are electronically ionized at 70 eV, followed by detection by a mass spectrometer as per their
*m/z*. HR-ToF-AMS can be operated in W-mode or V-mode. For this study, it was operated in V-mode with
a sampling time of 2 mins. Mass spectra (MS) mode, in which mass spectra of the components are
measured, and particle time-of-flight (PToF) mode in which the size-resolved mass spectra are measured,
are alternated in every 30s in 2 cycles. The HR-ToF-AMS was calibrated using standard protocols provided
in our previous publication (Lalchandani et al., 2021; Singh et al., 2019).
To determine the mass concentration of NR-PMs, Unit mass resolution (UMR) analysis was done using the
SQUIRREL data analysis toolkit (version 1.59) programmed in IGOR Pro 6.37 software (Wavemetrics,
Inc., Portland, OR, USA). High resolution (HR) analysis was also done on the data set using Peak Integrated
Key Analysis (PIKA version 1.19) toolkit. At the beginning of each campaign at the two sites, ionization
efficiency calibration was performed by injecting mono-disperse 300 nm ammonium nitrate and ammonium
sulphate particles into AMS and a condensation particle counter (Jayne et al., 2000). More details can be
found in Lalchandani et al. (2021) (manuscript under review).
**2.3.2 Heavy and trace metal measurements by ICP-MS and Xact 625i**
For the analysis of heavy and trace metals, 15 sq. cm area of each collected filter was digested in an acid
mix of 0.5 ml HF+1.5 ml $HNO_3$ for 4 hours within closed HDPE Teflon tubes using a Hot plate (Savillex-


HF resistive Model number 88888:00000). The temperature range should be ~90-120˚C to ensure complete
digestion of the elements. Further, 2.5 ml of $HClO_4$ was added to the precipitates, left over the Teflon tube
wall and the tubes were kept on the hot plate at 220-240˚C for another 4 hours with the lids open for
complete evaporation of the acid mix. Moreover, the residual was dissolved in 6N $HNO_3$ and diluted with
de-ionized water (resistivity-18.20 MΩ cm) followed by filtering through 0.22 μm quartz filter papers prior
analysis. Details can be found in our forthcoming paper. This method is well established and used in many
studies (Minguillón et al., 2012; Querol et al., 2008).
Thirty two metals (Al, Na, K, Ca, Ti, V, Cr, Mn, Fe, Ni, Cu, Zn, As, Se, Rb, Sr, Zr, Cd, Sn, Sb, Ba, Pb, Cs,
La, Ce, Pt, Tl, Mg, Li, Mo, Co and Pd) were analyzed for all filter extracts using an ICP-MS (Thermo
Scientific iCAP Q ICP-MS) at IIT Kanpur Environmental Engineering laboratory. Si could not be
determined in the filter samples because Si is the primary constituent of the quartz filters and hence digested
during sample preparation. Samples were first introduced to a nebulizer using an injector attached to an
autosampler for transformation into fine aerosol droplets followed by ionization at a very high temperature
(8000K) in Ar-plasma. The elements elute as per their *m/z*. A known concentration (5 ppb) of Ge was used
as an internal standard to monitor the instrumental drift during the analysis. The overall average drift was
reported as ± 10%. The calibration was performed by ten-point method with a range of mix standards
prepared by the serial dilution from the High purity multi-element (35 elements) standards (soluble in 1%
$HNO_3$, 100 ppm) purchased from Sigma Aldrich.
The Xact 625i Ambient Metals Monitor (Cooper Environmental Services (CES), Beaverton, OR, USA)
uses X-ray fluorescence to measure the real-time elemental data in particulate matter. For this study, a $PM_{2.5}$
inlet was used. Details of the instrument can be found in Furger et al., 2017. Briefly, aerosol samples were
collected on a Teflon filter tape followed by hitting the loaded area with X-rays and the fluorescence
measured by a silicon drift detector (SDD). Thirty elements: Al, Si, S, Cl, K, Ca, Ti, Cr, Mn, Fe, Co, Ni,
Cu, Zn, As, Se, Br, Rb, Sr, Zr, Mo, Cd, In, Sn, Sb, Te, Ba, Pb, Bi, and Bi were measured with 30 min time
resolution. The Xact 625i was calibrated during each campaign using thin film standards for the individual
elements. The reproducibility was observed within ± 5%. Every midnight energy alignment checks were
performed for 15 mins (00:15 to 00:30) (for Cr, Pb, Cd and Nb). An uncertainty of ~10% was reported by
the manufacturer in an interference free situation (USEPA & Etv, 2012). This included 1.73% from flow
(CEN, 2014), 5% for standard reproducibility or uncertainty during calibration (USEPA, 1999) and 2.9%
from term stability as reported in Tremper et al. (2018). More details on the instrumental set up and stability
check during the summer and winter campaign can be found in Rai et al. (2020) and Shukla et al. (2021).
**3. Results and discussions**



### 3.1. Online and offline measurements of WSIS and their comparison

Large temporal variability was observed in both offline and online measurements of WSIS ($NO_3^-$, $SO_4^{2-}$ and, $NH_4^+$ and $Cl^-$) during the winter campaign and summer campaign at both sites. The inorganics data with 2 mins interval from HR-ToF-AMS were averaged over the sampling period of the filters, i.e., 24 hours. The time series of $NO_3^-$, $SO_4^{2-}$, $NH_4^+$, and $Cl^-$ during the summer and winter campaign at IITD and winter campaign at IITMD are shown in Fig. SM1 in the supplementary material. Higher peaks of inorganics were observed during 25[th] Oct-18[th] Nov during the winter campaign at IITD, which was the agricultural crop-residue burning period (Nagar et al., 2017). During the winter campaign, $NO_3^-$ was the most abundant ion followed by $SO_4^{2-}$, $NH_4^+$ and $Cl^-$ for both offline and online measurements at both the sites whereas, during the summer campaign at IITD, $NO_3^-$ was the most abundant ion followed by $NH_4^+$, $SO_4^{2-}$ and $Cl^-$ in online measurement (HR-ToF-AMS). Similar results were observed in our companion paper, Shukla et al. (2021). Interestingly, in the case of offline measurements during the summer campaign at IITD, $NO_3^-$ was least abundant, and the sequence changed as $SO_4^{2-}> NH_4^+ >Cl^-> NO_3^-$. The average concentrations with their ranges are tabulated in Table SM1 in supplementary material, and the mean, maximum and minimum concentration are shown in Fig. 2 using box plots.

Comparability and correlation between offline and online measurements were evaluated in this study by applying linear regression using offline data as the independent and online data as the dependent variable. The comparability of $NH_4^+$ measurements was observed to be good for both summer and winter campaigns at both sites. During the winter campaign, the correlations were $R^2=0.76$, and $R^2=0.89$, and the slopes were closer to 1 (0.99 for IITD and 0.93 for IITMD) for IITD and IITMD, respectively (Fig. 3). Interestingly during the summer campaign at IITD, though the correlation improves ($R^2=0.91$), the slope decreases to 0.49. This is because the vapor concentration of sulfuric acid ($H_2SO_4$) is negligible during winters and higher during summers. As a result of which, the adsorption of sulfuric acid on PM deposited on the filter papers happens during summers which react with gaseous ammonia ($NH_3$) to form particulate ($NH_4$)$_2SO_4$ (D. Zhang et al., 2000), thus increasing ammonium concentration in the offline measurements during the warmer season, especially in the presence of dust (Nicolás et al., 2009).



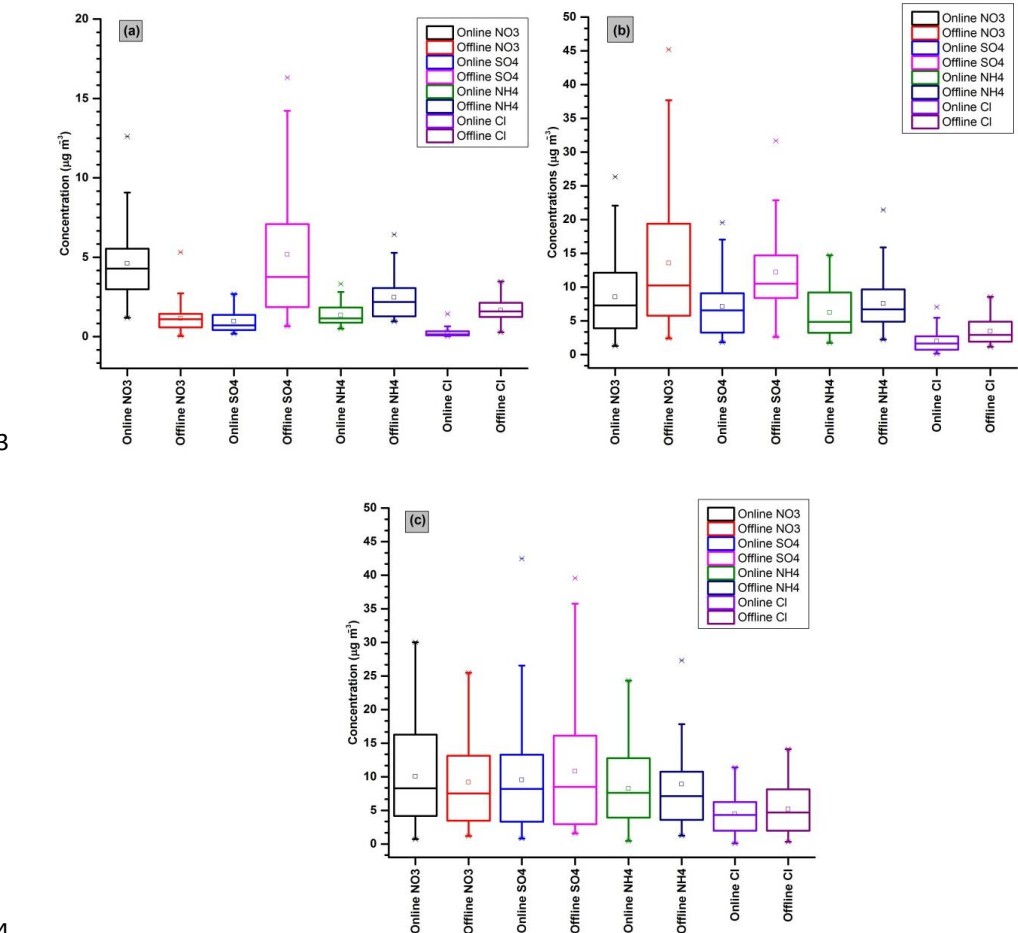

Fig.2. Box plots of online and offline measured secondary species ($NO_3^-$, $SO_4^{2-}$ and, $NH_4^+$) and $Cl^-$ during (a) summer campaign at IITD, (b) winter campaign at IITD, and (c) winter campaign at IITMD site.

The filter-based measurements of $SO_4^{2-}$ were higher than those from the online measurements for IITD and IITMD during both the seasons (Fig. 2). Their comparability is characterized by a correlation coefficient of $R^2=0.93$ with a slope of 0.17 and $R^2=0.82$ with a slope of 0.68 at IITD during the summer and winter campaigns, respectively. Interestingly, offline $SO_4^{2-}$ data correlates well with online $SO_4^{2-}$ data having a correlation coefficient of $R^2=0.93$ with a slope close to 1 (0.93) during the winter campaign at IITMD (Fig. 3). A slope of less than 0.5 was observed in Malaguti et al., (2015) in Italy during the warm period whereas the offline measurement of $SO_4^{2-}$ was 34% lesser than the AMS measurements in Pandolfi et al., (2014). The higher $SO_4^{2-}$ concentrations on the un-denuded offline filter-based measurements were possibly because of the positive sampling artifact. $SO_2$ is absorbed in the filter by the collected alkaline particles



(Nie et al., 2010). The higher concentration could also be due to the formation of particulate $(NH_4)_2SO_4$
because of the earlier discussed reactions between gas-phase $NH_3$ and $H_2SO_4$ formed on the fine particles
(Nicolás et al., 2009).

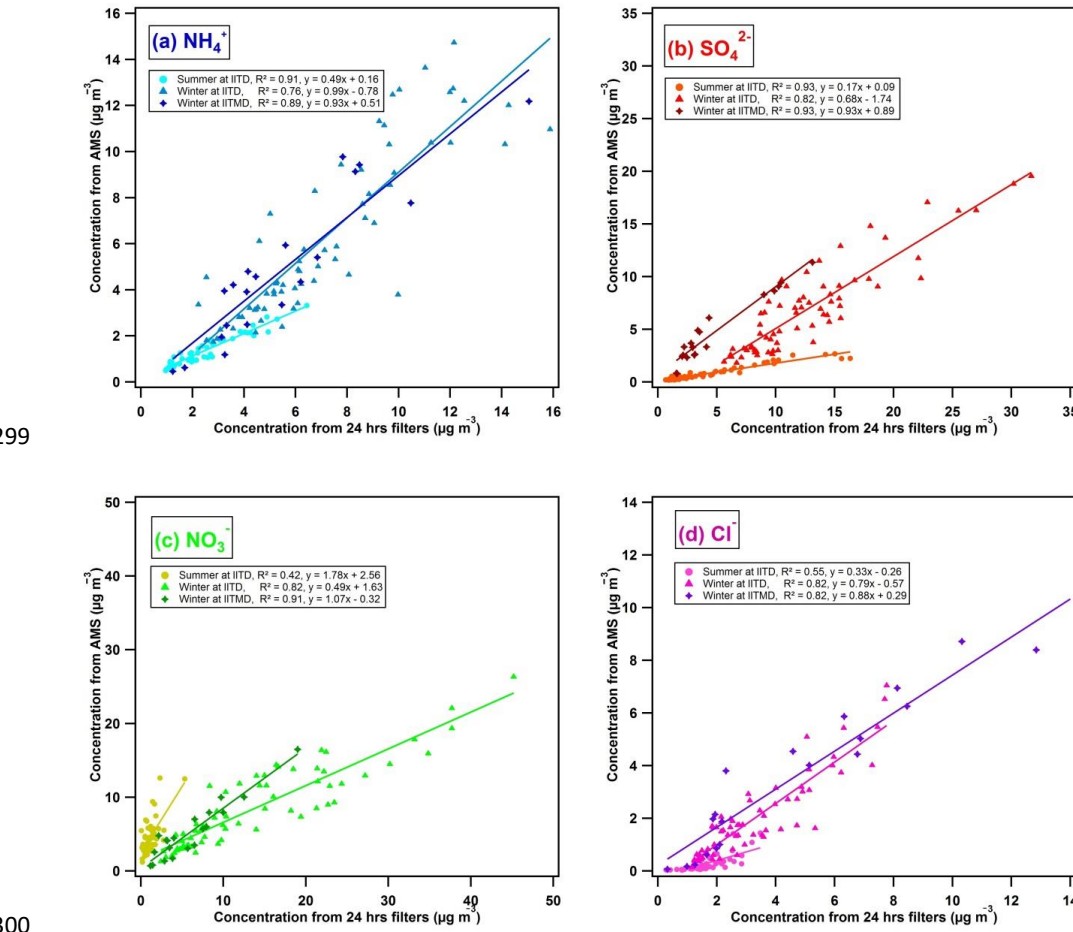





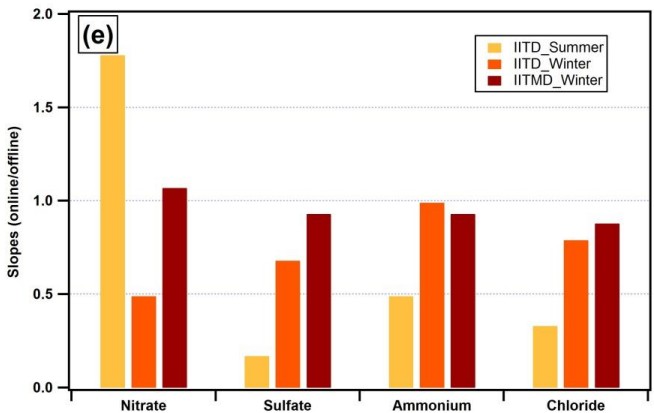

Fig.3. Scatter plots between online and offline measured (a) $NH_4^+$, (b) $SO_4^{2-}$, (c) $NO_3^-$, (d) $Cl^-$ concentrations and (e) comparison of slopes (online/offline) of the measured inorganic ions in $PM_{2.5}$ during summer and winter campaign at IITD and during winter campaign at IIITMD.

The $NO_3^-$ concentrations measured by the HR-ToF-AMS were higher than the offline data during summer at IITD and during winter at IITMD whereas, filter-based measurements of $NO_3^-$ were higher during winter at IITD (Fig. 2). The online and offline measurements posed a good correlation during winter ($R^2 = 0.91$ and slope of 1.07 at IITMD, $R^2 = 0.82$ and slope of 0.49 at IITD). The linearity worsens during summer at IITD ($R^2 = 0.42$ and slope of 1.78) (Fig. 3). The slopes and correlation coefficient for the WSIS are listed in Table 2. Pandolfi et al., (2014) observed $NO_3^-$_ HR-AMS/Filter ratios of ~1.7 at Barcelona and Montseny in Europe. The higher offline $NO_3^-$ concentrations during winter at IITD can be possibly because of the positive artifact due to the absorption of gas-phase nitric acid ($HNO_3$) on the filter (Chow, 1995). Many studies (Chow et al., 2008; Kuokka et al., 2007; Malaguti et al., 2015) reported higher concentrations of $NO_3^-$ from high time resolution measurements than filter-based measurements due to the evaporation of ammonium nitrate collected on filters over the duration of sample collection (Pakkanen & Hillamo, 2002; Schaap et al., 2004; Kuokka et al., 2007). This evaporation loss increases with the decrease of humidity and the increase of temperature (Chow et al., 2008; Takahama et al., 2004). Also, complete evaporation may occur beyond 25°C (Schaap et al., 2004). The high temperature (35°C-48°C) during the long sampling hours (24 hours) may be a possible reason for the poor correlation between online and offline $NO_3^-$ measurements during the summer campaign at IITD. Chow et al., (2008) observed the evaporation loss from quartz filter to be more than 80% during the warm season in central California. A study by Schaap et al., (2004) reported that the $NO_3^-$ volatilization during a 24-h sampling period not only depends on the sampling apparatus and ambient conditions, but also a function of sampling strategy. If the sampling





strategy is evening to evening (24 hours), the samples will lose the $NO_3^-$ during night as evaporation
increases during the day with the increase in temperature. However, during morning-to-morning strategy,
the filters will collect the $NO_3^-$ at night, and the higher temperature in the afternoon of the previous day
may promote the loss of $NO_3^-$ from the filter (Malaguti et al., 2015). In this study, the sampling time was
from 6:30 am to the next day 6.30 am. Therefore, the filter-based inorganic measurements suffered from a
negative sampling artifact due to the evaporation of nitrate collected during the forenoon in a temperature
of 20°C-25°C during the winter campaign and ~ 38°C-45°C during the summer campaign.
Table 2. Regression coefficients and slopes for the comparison of WSIS measured by HR-ToF-AMS and
IC.

| Sites | $NO_3^-$ | | $SO_4^{2-}$ | | $NH_4^+$ | | $Cl^-$ | |
|---|---|---|---|---|---|---|---|---|
| | $R^2$ | Slope | $R^2$ | Slope | $R^2$ | Slope | $R^2$ | Slope |
| **IITD Summer** | 0.42 | 1.78 | 0.93 | 0.17 | 0.91 | 0.49 | 0.55 | 0.33 |
| **IITD Winter** | 0.82 | 0.49 | 0.82 | 0.68 | 0.76 | 0.99 | 0.82 | 0.79 |
| **IITMD Winter** | 0.91 | 1.07 | 0.93 | 0.93 | 0.89 | 0.93 | 0.82 | 0.88 |

We observed higher $Cl^-$ concentration in filter-based measurement than online measurement using HR-ToF-
AMS during both campaigns at IITD and winter at IITMD. A good correlation of $R^2= 0.82$ with a slope of
0.79 and $R^2= 0.82$ with a slope of 0.88 was observed during the winter campaign at IITD and IITMD,
respectively (Fig. 3). Interestingly, during the summer campaign at IITD the comparability was moderate
with a correlation coefficient of $R^2= 0.55$ and a slope of 0.33. A correlation coefficient of $R^2= 0.83$ between
$Dp <10$ μm measured with the MARGA and analyzed from Teflon filters was reported in Makkonen et al.,
(2012) during Feb-May. We also compared $Cl^-$ measurements from Xact 625i with the measurements from
IC. IC measurements of $Cl^-$ were also found to be higher than Xact 625i measurements during summer at
IITD and winter at IITMD. Interestingly, the $Cl^-$ measurements from Xact 625i were ~1.9 times higher than
the measurements from IC during winter at IITD (see fig. SM2 in supplementary material). The correlations
were found to be good during winter ($R^2= 0.83$ and 0.76 at IITD and IITMD respectively) and worsen
during summer ($R^2= 0.27$ at IITD), similar to what we observed for AMS_$Cl^-$ and IC_ $Cl^-$. Lower
temperature and higher RH during winter retains $Cl^-$ in particulate phase for long enough to be detected
which is not the case in summer. Further, $Cl^-$ is predominantly found in the coarse fraction. Also, while
AMS only measures the NR-$Cl^-$ (Manchanda et al., 2021), e.g. $NH_4Cl$, which can vaporize at 600°C but
cannot measure $Cl^-$ from refractory-KCl, IC measures chloride from all the water-soluble chloride salts,
including $NH_4Cl$ and KCl. On the other hand, Xact 625i measures $Cl^-$ from both the salts using XRF
technique. This probably justifies the lower concentration of $Cl^-$ in online measurements than filter-based
measurements and better slopes (closer to unity) with Xact than AMS.





### 3.2 Online and offline measurements of heavy and trace metals and their comparison

For the inter-comparison of heavy and trace metal concentrations from Xact 625i and ICP-MS, the half-hourly Xact 625i data were averaged to 24 hours filter sampling interval. A total of 32 elements were analyzed on each filter using ICP-MS while, 27 elements were measured in Xact 625i at IITD during summer and winter campaign, and 30 elements were measured in Xact 625i at IITMD during the winter campaign. The spatial and temporal variations of the crustal and trace elements are shown in Fig. SM3. Al and Ca concentrations were the most abundant in ICP-MS and Xact measurements respectively during the summer season (Fig. 4a, 5a & 5b) because of the increase in the crustal activities whereas K concentrations were significantly high for both the measurements during winter (Fig. 4d, 4g, 5c, 5d, 5e and 5f) due to mass-scale agricultural crop-residue burning in the adjoining states of Punjab and Haryana. Elements emitted from anthropogenic activities, e.g., coal-fired power plants (As, Se, Hg, Pb), traffic emissions (Cr, Pb, Mn), wear debris emission (Cu, Cd, Fe, Ga, Mn, Mo), etc. are found to be in higher concentration during winter campaign than summer campaign for both the measurements. Similar results were observed in our companion paper Shukla et al. (2021). The average values of metals along with their ranges are tabulated in Table SM2 and the statistics involved mean, upper, and lower values are shown in Fig. 4.

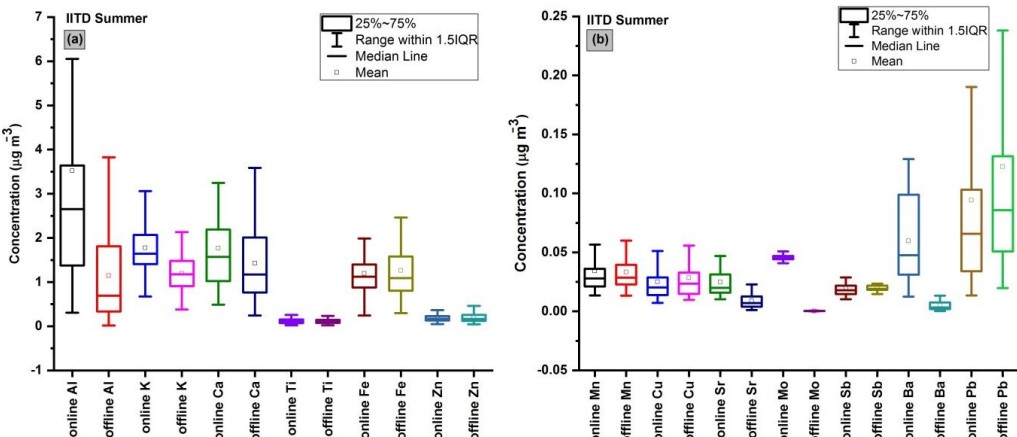










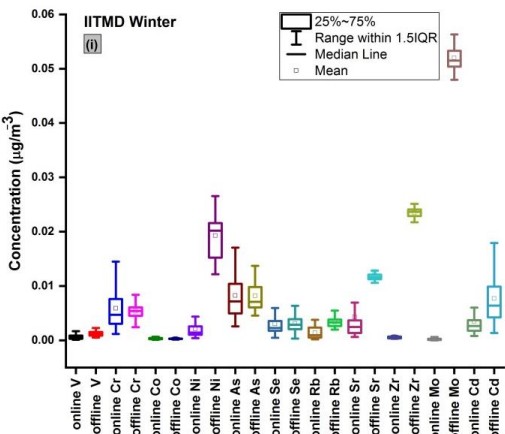


Fig.4. Box plots of online and offline measured heavy and trace metals during (a,b,c) summer campaign at
IITD, (d,e,f) winter campaign at IITD, and (g,h,i) winter campaign at IITMD site.
The trends of the elemental concentration in decreasing order for both ICP-MS and Xact measurements
during summer and winter at IITD and winter at IITMD are shown in Fig. SM4. The trace metals such as
Cd, Mn, Mo, Ba, Pd etc. contribute a small portion in $PM_{2.5}$ in terms of their mass concentration but have a
significant effect on human health. Fractions of elements in total element concentration for both the
measurements during summer and winter campaign at IITD and during winter campaign at IITMD were
shown in Fig. 5.

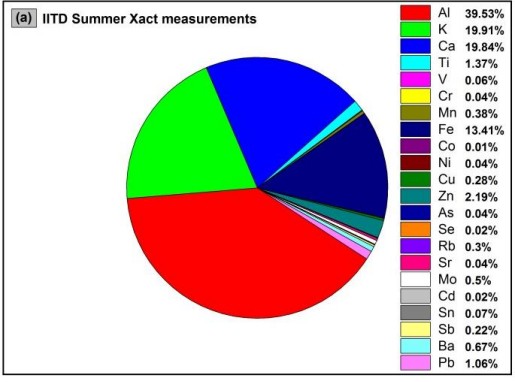

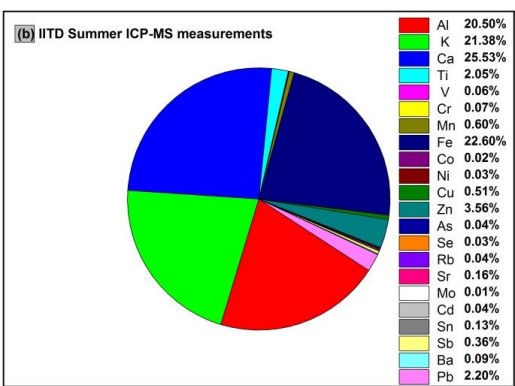




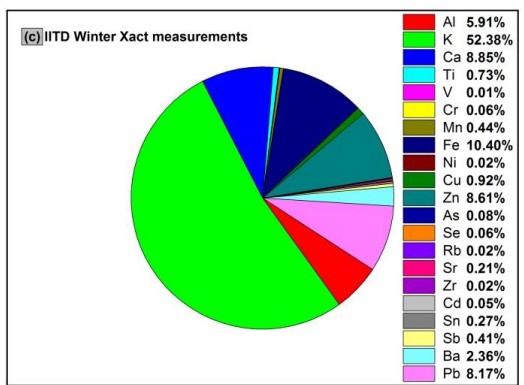

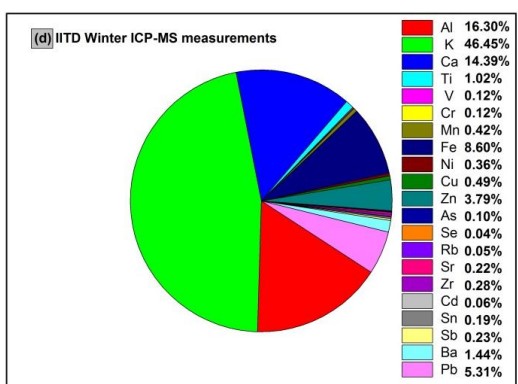


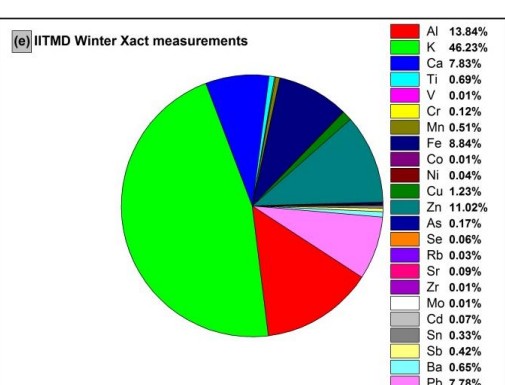

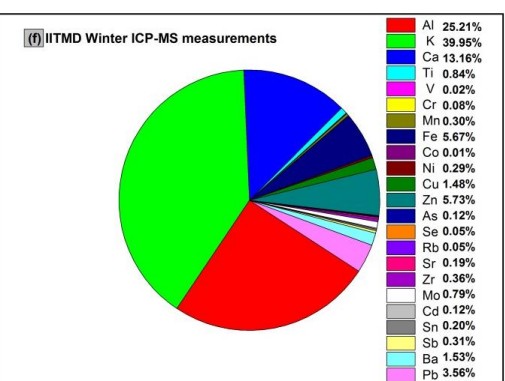


Fig.5. Fractions (%) of elements in total element concentration in PM$_{2.5}$ presented in pie format for online (a,c,e) and offline (b,d,f) measurements during winter and summer at IITD and during winter at IITMD.

MDLs for ICP-MS measurements were calculated according to Escrig Vidal et al., (2009), and MDLs for Xact 625i were obtained from the manufacturer. MDLs for the Xact 625i and ICP-MS are listed in Table SM2 in the supplementary material. Though the half-hourly Xact data were averaged to 24 hours to the corresponding interval of the filter sampling, for comparability check, MDLs of Xact 625i measurements were taken for 30 mins sampling time while MDLs of filter-based elemental measurements were calculated for 24 hours. Data below 3 × MDL value were discarded from the data set as it would lead to higher uncertainty (Furger et al., 2017). The elements K, Ca, Ti, Mn, Fe, Ba, and Pb have >80% of their values above both offline and online MDLs, and thus the data quality is reliable. Further, Ni, Mo, Zr have higher blank concentrations, and thus the data is not reliable for ICP-MS measurements. The comparability of the elements measured online using Xact 625i with those analysed using an ICP-MS, was checked for the common elements in these two measurements (21 elements for IITD and 23 elements for IITMD) and is shown in Fig. 6.










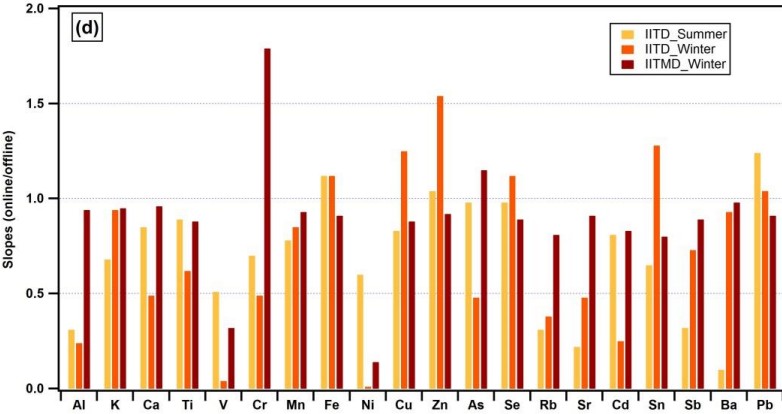


Fig. 6. Scatter plots and regression lines of Xact 625i vs. ICP-MS data for groups A, B, and C during (a)
summer at IITD, (b) winter at IITD, (c) winter at IITMD, and (d) comparison of slopes (online/offline) of
the measured heavy and trace metals in PM$_{2.5}$ during the summer and winter campaign at IITD and winter
campaign at IIITMD.

Based on their comparability characteristics, elements were grouped under 3 categories. Group A showed
excellent linearity between the two methods with a correlation coefficient of $R^2 > 0.8$. Overall, group A
consists of Al, K, Ca, Ti, Mn, Fe, Cu, Zn, Ba, and Pb during winter at IITD whereas, Cu showed in another
group during summer at IITD. Though Sr, Cd, Sn, Sb had average values below MDLs and posed a lower
correlation coefficient at IITD, interestingly, Sr, Cd, Sn, and Sb joined group A during winter at IITMD.
To distinguish the potential difference in accuracy between the two methods, intercepts were not forced to
be zero. The slopes are important, which indicate biases between the two measurements. The slopes of Zn,
Fe, and Pb were closer to unity during summer at IITD (Fig. 6). K, Fe, Cu, Zn, Ba, and Pb achieved a slope
of 0.94-1.25 during winter at IITD whereas, Mn, Fe, Zn, Pb, Sr, Sn, and Sb pose a slope slightly higher
than unity during winter at IITMD (Fig. 6). A slight difference in cut-off value for the particle size can
reduce the slopes from unity and produce ~10% difference in collected mass (Panteliadis et al., 2012). The
slopes and the correlation coefficients are listed in Table 3.

In a comparison study conducted by USEPA & Etv. (2012) between Xact 625i and ICP-MS measurements,
Ca, Cu, Mn, Pb, Se, and Zn were highly correlated except Cu. Cu was close to the MDL values of ICP-MS
and Xact 625i. A good agreement was observed between Xact 625i and offline measurements using ED-
XRF in South Korea by Park et al., (2014). The comparability between Xact measurements and ICP-MS
measurements was checked for the elements As, Ba, Ca, Cr, Cu, Fe, K, Mn, Ni, Pb, Se, Sr, Ti, V, and Zn



in Tremper et al. (2018). They observed an average $R^2$ of 0.93 and a slope of 1.07 for these elements. In the
study by Furger et al., (2017) during a warmer season in Switzerland, an excellent correlation ($R^2>0.95$)
was found for Xact and ICP-OES/MS measurements of S, K, Ca, Ti, Mn, Fe, Cu, Zn, Ba, and Pb. However,
they found that the elemental measurements by Xact 625i were 28% higher than ICP-OES/MS
measurements for these 10 elements. In our study, Xact measurements of Fe, Cu, Zn, and Pb yielded an
average of 24% higher mass concentrations than ICP-MS measurements for Group A elements during
winter at IITD. Xact measurements were systematically 10% (average) higher than ICP for Zn, Fe, and Pb
during summer at IITD, whereas;  we obtained an average of 41% higher Xact measurements than ICP for
Mn, Fe, Zn, Pb, Sr, Sn, and Sb during winter at IITMD in Group A elements.
Group B was characterized by moderate linearity having $R^2\sim0.4\text{-}0.8$ and consisted of the elements V, Cu,
As, and Se during summer at IITD; V, Cr, As, Se, Sn during winter at IITD and Cr, As and Se during winter
at IITMD. These elements in Group B had their values very close to or below  MDLs of at least one of the
analysis methods. During winter at both sites, Cr and As from ICP-MS have ~50-65% of their values below
MDLs, whereas ~68-72% of their values were above the MDLs of Xact 625i. Though some of the elements
in this group in different seasons have slopes near or greater than unity (see Table 3) like Group A, their
comparison is not statistically feasible.
Table 3. Regression coefficients and slopes for the comparison of Xact 625i and ICP-MS measurements.

| Sites | Group A | Slope | $R^2$ | Group B | Slope | $R^2$ | Group C | Slope | $R^2$ |
|---|---|---|---|---|---|---|---|---|---|
|  | Al | 0.31 | 0.81 | V | 0.51 | 0.59 | Cr | 0.7 | 0.29 |
|  | K | 0.68 | 0.93 | Cu | 0.83 | 0.70 | Co | - | - |
|  | Ca | 0.85 | 0.85 | As | 0.98 | 0.62 | Rb | 0.31 | 0.26 |
| **IITD Summer** | Zn | 1.04 | 0.83 | Se | 0.98 | 0.77 | Sr | 0.22 | 0.24 |
|  | Ba | 0.1 | 0.94 |  |  |  | Mo | - | - |
|  | Ti | 0.89 | 0.92 |  |  |  | Cd | 0.81 | 0.42 |
|  | Mn | 0.78 | 0.92 |  |  |  | Sn | 0.65 | 0.41 |
|  | Fe | 1.12 | 0.93 |  |  |  | Sb | 0.32 | 0.38 |
|  | Pb | 1.24 | 0.95 |  |  |  | Ni | 0.6 | 0.4 |
|  | Al | 0.24 | 0.89 | V | 0.04 | 0.48 | Ni | 0.01 | 0.07 |
|  | K | 0.94 | 0.85 | Cr | 0.49 | 0.67 | Rb | 0.38 | 0.24 |
|  | Ca | 0.49 | 0.89 | As | 0.48 | 0.42 | Sr | 0.48 | 0.34 |
|  | Ti | 0.62 | 0.89 | Se | 1.12 | 0.75 | Zr | - | - |





| | | | | | | | | | |
|---|---|---|---|---|---|---|---|---|---|
| **IITD Winter** | Mn | 0.85 | 0.91 | Sn | 1.28 | 0.53 | Cd | 0.25 | 0.16 |
| | Fe | 1.12 | 0.81 | | | | Sb | 0.73 | 0.40 |
| | Cu | 1.25 | 0.97 | | | | | | |
| | Zn | 1.54 | 0.98 | | | | | | |
| | Ba | 0.93 | 0.96 | | | | | | |
| | Pb | 1.04 | 0.95 | | | | | | |
| | Al | 0.94 | 0.39 | Cr | 1.79 | 0.65 | Ni | 0.14 | 0.13 |
| | K | 0.95 | 0.91 | As | 1.15 | 0.66 | Rb | 0.81 | 0.39 |
| | Ca | 0.96 | 0.45 | Se | 0.89 | 0.5 | Mo | - | - |
| | Ti | 0.88 | 0.72 | | | | Zr | - | - |
| | Mn | 0.93 | 1.71 | | | | V | 0.32 | 0.37 |
| | Fe | 0.91 | 1.29 | | | | Co | 0.36 | 0.04 |
| | Cu | 0.88 | 0.70 | | | | | | |
| **IITMD Winter** | Zn | 0.92 | 1.24 | | | | | | |
| | Ba | 0.98 | 0.36 | | | | | | |
| | Pb | 0.91 | 1.41 | | | | | | |
| | Sr | 0.91 | 1.53 | | | | | | |
| | Cd | 0.83 | 0.47 | | | | | | |
| | Sn | 0.8 | 1.44 | | | | | | |
| | Sb | 0.89 | 1.23 | | | | | | |

The Group C elements, e.g., Ni, Rb, Sr, Zr, Cd, and Sb during summer at IITD; Cr, Co, Rb, Sr, Mo, Cd,
Sn, Sb, and Ni during winter at IITD and, Ni, Rb, Mo, Zr, V, and Co during winter at IITMD are
characterized by their bad correlation ($R^2<0.4$). Interestingly, measurements of some element e.g. Mo
during summer and winter at IITD and IITMD respectively, Co during summer at IITD, and Zr during
winter at both sites from both the methods did not correlate at all. For most of the elements in this group,
70-85% of measurements were below both method's MDLs and rest of the data were below $3 \times$ MDLs.
The high and variable blank concentrations of these elements increased the MDL values in ICP-MS
measurement. The particle size dependent self-absorption effect and line interference between different
elements in Xact measurement could also increase the MDL values (Furger et al., 2017). This is probably
the reason for the values lower than MDLs for the elements in this group.
Overall, we observed 10-40% higher Xact measurements than ICP for some of the elements in Group A
during different seasons. The difference in the Xact and High-volume sampler inlet location and their
distance from the road can cause such difference in measurements (Furger et al., 2017). In the case of dust
resuspension from vehicular traffic, the number concentration of finer particulate matter tends to decrease
sharply within an increment of just 50m from the roadway (Hagler et al., 2009). In this study, we tried to
co-locate the two sample inlets, but it could not be avoided. Also, IITD and IITMD are both very close
(<200 m) to a roadway with moderate to heavy-duty traffic. The differences in online and offline
measurements may indicate a gradient in some elements due to very close proximity to heavy-duty traffic.
Also, the different temperatures of the sample inlets may give rise to a difference in measured
concentrations from both methods (Tremper et al., 2018). In this study, the blank corrected ICP-MS
measurements may result in overestimation or underestimation due to variable and high blank
concentration. The difference can also occur due to the digestion recovery rate for the digestion protocol
used for the filter analysis. Moreover, if the ambient elemental concentration is much lower than the
standards used for calibration of Xact, such differences may occur (Indresand et al., 2013).
**4. Conclusion**
Atmospheric WSIS ($NO_3^-$, $SO_4^{2-}$, $NH_4^+$, $Cl^-$) and heavy and trace elements in $PM_{2.5}$ were measured using
offline methods (IC for WSIS and ICP-MS for elements) and online methods (HR-ToF-AMS for inorganics
and Xact 625i for elements). These measurements were compared to assess the measurement quality and
sampling artifacts of these measurement techniques in the heavily polluted Delhi-NCR for two different
metrological conditions (winter and summer season). Field campaigns were conducted at two NCR sites,
namely, IITD during June-July 2019 and Oct-Dec 2019 and at IITMD during Oct-Dec 2019. The key
findings of this study are summarized below.
- $NH_4^+$ concentrations from the IC and HR-ToF-AMS, compared well during winter with a slope of
0.99 at IITD and 0.93 at IITMD. Interestingly, $NH_4^+$ concentrations were higher in offline
measurements during summer at IITD. The decrease in slope was probably due to the formation
of particulate $(NH_4)_2SO_4$.
- Offline $SO_4^{2-}$ measurements were higher (with a slope of 0.17 during summer at IITD and 0.8 and
0.93 during winter at IITD and IITMD, respectively) during both seasons at both the sites due to
the positive sampling artifact. The absorption of $SO_2$ and the oxidation or condensation process
may result in additional sulphate.
- Lower $NO_3^-$ concentrations (with a slope of 1.78 during summer at IITD and 1.07 during winter
at IITMD) were observed in the offline measurement during summer at IITD and during winter
at IITMD because of the evaporation of $NH_4NO_3$ from the filter substrates. The evaporative loss
of nitrate from the filters was minimal in winter at IITMD. It aggravated during summer at IITD





due to the evaporation of ammonium nitrate in such a high temperature (35°C-48°C) range. The
higher $NO_3^-$ concentrations (slope~0.49) in the filters than HR-ToF-AMS measurements during
winter at IITD can be due to the absorption of gas-phase $HNO_3$ on the filter.
• Offline $Cl^-$ was consistently higher (with a slope of 0.33 during summer at IITD and 0.79 and 0.88
during winter at IITD and IITMD, respectively) than HR-ToF-AMS measurements during both
seasons at both sites mainly because HR-ToF-AMS only measures the $NR-Cl^-$ whereas, the offline
$Cl^-$ measurements includes chloride from all the water-soluble chloride salts. The comparability
degrades during summer due to the volatile nature of $Cl^-$ in higher temperatures and lower RH.
• The elements were grouped into three categories (Group A, B and C) according to their
comparability characteristics. The elemental data from Xact 625i were highly correlated ($R^2$>0.8)
with ICP-MS measurements of the 24 hr filters for Group A elements (Al, K, Ca, Ti, Zn, Mn, Fe,
Ba, and Pb). The Cu also showed up in this group during winter at IITD. About 80% of the data
for these elements were above MDLs for both the methods. Though Sn, Sb, and Cd had values
below MDLs of one or both the methods, interestingly, they were highly correlated ($R^2$>0.8) and,
slopes are very close to unity for Sn and Sb during winter at IITMD. The correlation coefficients
> 0.8 for the elements in Group A indicated the high precision of the online and offline
measurements. Hence, these elements from any of these methods can be reliably used for
modelling studies.
• The elements under Group B had their values closer to or below at least one of the method's
MDLs. Cr and As had ~50-65% of their values below ICP-MS MDLs from ICP-MS, whereas
~68-72% of their values were above Xact 625i MDLs during winter at both sites.
• Elements like Ni, Mo, and Zr measured from ICP-MS were not reliable due to their higher and
variable blank concentrations. No conclusion on their measurement accuracy by the two methods
can be drawn.
• In summary, the daily averaged half-hourly Xact 625i measurements were 10-40% higher than 24
hr filter measurements by ICP-MS depending upon the seasons, sites and elements in Group A.
Distance between two inlets for the two methods, distance of the inlets from the roadway, line
interference between two elements in Xact measurements, particle size, sampling strategy, filter
type, higher and variable concentrations in blank filters and digestion protocol for ICP
measurements can cause the difference in measurements between the two methods.
The above findings highlight the measurement methods' accuracy and implement the particular type of
measurements as needed. Future work should involve using different filter substrates and different
digestion protocols to reevaluate the difference between these online and offline methods. Although this



study compares the PM species, a comparison of full source apportionment analysis between online and
offline methods should be done for more qualitative and quantitative insights.
**Author contribution**
HSB performed the offline analysis, data processing and wrote the manuscript. JD collected the AMS data
at IITD. AS and VL carried out the Xact and AMS data collection and processing respectively. NR, MK,
VS and SNT were involved with the supervision and conceptualization. All co-authors contributed to the
paper discussion and revision.
**Declaration of interests**
The authors declare that they have no conflict of interest.
**Acknowledgements**
The authors are thankful to Naba Hazarika and Mohd Faisal for collecting the Xact data at IITMD and
Pawan Vats for the sampling and collection of the filters at IITD. The authors are also thankful to Amit
Vishwakarma, Vaibhav Shrivastava and Harishankar for helping in offline analysis. This work is financially
supported by the Department of Biotechnology, Government of India (Grant No.
BT/IN/UK/APHH/41/KB/2016-17 dated 19[th] July 2017) and Central Pollution Control Board (CPCB),
Government of India to conduct this research under grant no. AQM/Source apportionment_EPC
Project/2017 dated 12[th] February 2019.

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
