# Peer review of "Inter-comparison of online and offline methods for measuring ambient heavy and trace elements and water-soluble inorganic ions (NO3, SO42, NH4 and Cl ) in PM2.5 over a heavily polluted 2 3 megacity, Delhi 4 Himadri Sekhar Bhowmik1,"

_Atmospheric Measurement Techniques, 2021_

## Author Comment (AC1)

**Inter-comparison of online and offline methods for measuring ambient heavy and trace elements and water-soluble inorganic ions (NO3-, SO42-, NH4+ and Cl-) in PM2.5 over a heavily polluted megacity, Delhi**

Himadri Sekhar Bhowmik1, Ashutosh Shukla1, Vipul Lalchandani1, Jay Dave2, Neeraj Rastogi2, Mayank Kumar3, Vikram Singh4, and Sachchida Nand Tripathi1\*

1 Department of Civil Engineering, Indian Institute of Technology Kanpur, Kanpur, India

1\* Department of Civil Engineering and Centre for Environmental Science and Engineering, Indian Institute of Technology Kanpur, Kanpur, India

2 Geosciences Division, Physical Research Laboratory, Ahmedabad, India

3Department of Mechanical Engineering, Indian Institute of Technology Delhi, New Delhi, India

4Department of Chemical Engineering, Indian Institute of Technology Delhi, New Delhi, India

Correspondence to: S. N. Tripathi (snt@iitk.ac.in)

**Responses (text in blue) to comments by the reviewer (text in black)**

We thank the referee for his/her valuable comments which have greatly helped us to improve the manuscript. Please find below our point-by-point responses (in blue) after the referee comments (in black). The changes or existing lines in the revised manuscript are written in *italic (red)*.

**Referee #1**

**General Comments:** The authors compare online and offline methods for the determination of elements and ions at highly polluted sites in Delhi. The topic is important and online methods would be needed for receptor models as they have better time-resolution. However, online instruments have not been very comparable with the traditional methods, there are problems e.g. with calibration and detection limits.

1. 32 The higher concentrations on the filters were due to the formation of particulate  $(NH_4)_2SO_4$ . This needs clarification (why?).

Reply: To keep the abstract short and precise we didn't explained the details. Though it has been explained in line 294-299 of the MS as-

"Interestingly during the summer campaign at IITD, though the correlation improves ( $R^2=0.91$ ), the slope decreases to 0.49. This is likely because the vapor concentration of sulfuric acid ( $H_2SO_4$ ) is higher during summers. As a result of which, the adsorption of sulfuric acid on PM deposited on the filter papers happens during summers which reacts with gaseous ammonia (NH3) to form a relatively stable particulate (NH4)2SO4 (Zhang et al., 2000), thus increasing ammonium concentration in the offline measurements during the warmer season, especially in the presence of dust (Nicolás et al., 2009)."

And in line 311-318 of the MS as-

"The higher  $SO_4^{2^2}$  concentrations on the un-denuded offline filter-based measurements could be due to refractory sulfate (e.g., potassium or calcium sulfate). The higher filter-sulfate could also be possibly because of the positive sampling artifact. The  $SO_2$  is absorbed on the filter by the collected alkaline particles (Nie et al., 2010). The higher concentration could also be due to the formation of ammonium bi-sulfate or ammonium sulfate because of the reaction between gas-phase ammonia with the acidic aerosols (Nicolás et al., 2009). Also, the un-denuded filter measurements could lead to higher filter-sulfate. The interaction of gas phase ammonia with acidic aerosols can be minimized by using denuders while collecting aerosols on the filters (Nault et al., 2020)."

2. 34 Why offline Cl- was higher than that measured by AMS?

Reply: Again we tried to keep the abstract short and precise by avoiding detailed explanations which were discussed in the section 3.1. It has been explained in line 366-368 of the MS as-

"Also, while AMS only measures the NR-Cl- (Manchanda et al., 2021), e.g. NH4Cl, which can vaporize at 600°C but cannot measure Cl- from refractory-KCl, IC measures chloride from all the water-soluble chloride salts, including NH4Cl and KCl."

3.82-83... 40 elements, e.g.--- there are more than 40 elements in the list?

Reply: We thank the reviewer for pointing this out. We mistakenly wrote 40. We rectified this in the line 85-87 in the MS as-

"It is capable of measuring 45 elements, e.g., Al, Si, P, S, Cl, K, Ca, Ti, V, Cr, Mn, Fe, Co, Ni, Cu, Zn, Ga, Ge, As, Se, Br, Rb, Sr, Y, Zr, Nb, Mo, Pd, Ag, Cd, In, Sn, Sb, Te, I, Cs, Ba, La, Ce, Pt, Au, Hg, Tl, Pb and Bi with a frequency of every 30 mins to 4 hours."

4. p.5 What were the cut-offs of each instrument? How about inlet lengths and materials?

Reply: The cut-offs of both HR-ToF-AMS and Xact 625 was 2.5µm.

For HR-ToF-AMS, at IITD the inlet length was 2.44 meter and it was made of stainless steel having inner diameter of 0.3 inch and outer diameter of 0.4 inch whereas, at IITMD, a 1.5 m long silicon tubing (TSI Inc) having 0.19 inch inner diameter was used as inlet. It was mentioned in the line no 169-173 in the MS as-

"Ambient fine particulate matters were sampled through  $PM_{2.5}$  cyclone (BGI, Mesa Labs. Inc.) inlet at IITD with a flow rate of 5 lpm (l/min) using a 2.44 m long stainless-steel tubing (0.3 inch I.D and 0.4 inch O.D) and through black silicon tubing (0.19 inch I.D) at IITMD, placed 1.5 m above the rooftop. A Nafion dryer (MD-110-144P-4: Perma Pure, Halma, UK) was used to dry the ambient aerosols to maintain the output RH at 20%."

For Xact 625, at both the sites, the inlet length was 2.44 meter and it was made of aluminium having inner diameter of 1.25 inch. It was mentioned in the line no 180-182 in the MS as-

"A separate sampling line of 2.44 meter (1.25 inch I.D) for the Xact which was made of aluminium was installed. A heater was set up at the end of the sampling line to ensure 45% RH set point."

5. 6-7. What was number of samples for each measurement period?

Reply: Daily 24 h PM2.5 samples were collected during both the measurement periods. A total of 64 filters (60 from IITD site during June-July 2019 including 4 blanks) were collected in summer whereas, a total of 186 filters (90 from each site during October-December 2019 plus 6 blanks) were collected in winter. It was added in the line no 155-157 in the MS as-

"A total of 64 filters (60 from IITD site during June-July 2019 including 4 blanks) were collected in summer whereas, a total of 186 filters (90 from each site during October-December 2019 plus 6 blanks) were collected in winter."

6. p.9 USEPA written here is as US-EPA and Usepa in the reference list.

Reply: We thank the reviewer for pointing out this mistake. The same has been corrected in the revised manuscript.

7. p.10 D. Zhang should be without D.

Reply: We thank the reviewer for pointing out this mistake. The same has been corrected in the revised manuscript.

8. p.13  $NO_3^-$  Why the filter method gave higher  $NO_3^-$  compared to online method in winter? The higher offline  $NO_3^-$  concentrations during winter at IITD can be possibly because of the positive artifact due to the absorption of gas-phase nitric acid (HNO3) on the filter. Why not in summer?

Reply: At IITD, the filter method gave higher  $NO_3^-$  compared to online method in winter likely due to the absorption of gas-phase nitric acid (HNO3) on the filter.

This is not the case in summer. During summer, the evaporation of ammonium nitrate collected on filters can be much higher than the absorption of gas-phase nitric acid. The evaporation increases with the decrease of humidity and the increase of temperature (Chow et al., 2008; Takahama et al., 2004). Also, complete evaporation may occur beyond  $25^{\circ}C$  (Schaap et al., 2004). Chow et al., (2008) observed the evaporation loss from quartz filter to be more than 80% during the warm season in central California. The high temperature ( $35^{\circ}C-48^{\circ}C$ ) during the long sampling hours (24 hours) may be a possible reason for the poor correlation between online and offline  $NO_3^-$  measurements during the summer campaign at IITD.

9.p.13 This part is not very logically written and difficult to follow: first summer results are compared, then winter and back to summer conditions and evaporation problem.

Reply: We have re-written the part in a logical way which is now easier to follow in line 329-346 in MS as-

"The online and offline NO3 measurements posed a good correlation during winter ( $R^2 = 0.91$  and slope of 1.07 at IITMD,  $R^2 = 0.82$  and slope of 0.49 at IITD) whereas the correlation worsens during summer at IITD ( $R^2 =$ 0.42 and slope of 1.78) (Fig. 3). The slopes and correlation coefficient for the WSIS are listed in Table 2. The  $NO_3$  concentrations measured by the HR-ToF-AMS were higher than the offline data during summer at IITD and during winter at IITMD whereas, filter-based measurements of  $NO_3^-$  were higher during winter at IITD (Fig. 2). The higher offline  $NO_3^-$  concentrations during winter at IITD can be possibly because of the positive artifact due to the absorption of gas-phase nitric acid (HNO3) on the filter (Chow, 1995). Many studies (Chow et al., 2008; Kuokkaet al., 2007; Malaguti et al., 2015) reported higher concentrations of  $NO_3^-$  from high time resolution measurements than filter-based measurements due to the evaporation of ammonium nitrate collected on filters over the duration of sample collection (Pakkanen & Hillamo, 2002; Schaap et al., 2004; Kuokkaet al., 2007). Pandolfi et al. (2014) observed  $NO_3^-$  HR-AMS/Filter ratios of ~1.7 at Barcelona and Montseny in Europe. This evaporation loss increases with the decrease of humidity and the increase of temperature (Chow et al., 2008; Takahama et al., 2004). Also, complete evaporation may occur beyond 25°C (Schaap et al., 2004). Chow et al., (2008) observed the evaporation loss from quartz filter to be more than 80% during the warm season in central California. The high temperature  $(35^{\circ}C-48^{\circ}C)$  during the long sampling hours (24 hours) may be a possible reason for the poor correlation between online and offline  $NO_3^-$  measurements during the summer campaign at IITD."

10. 323-327 This part should be rewritten:  $NO_3^-$  will not be lost during night, at night filters collect more quantitatively that during warmer daytime...

Reply: We have re-written this part in line 346-352 in MS as-

"Schaap et al. (2004) reported that the  $NO_3^-$  volatilization during a 24-h sampling period not only depends on the sampling instruments and ambient conditions, but also on sampling strategy. If the sampling strategy is evening to evening (24 hours), the samples will lose the  $NO_3^-$  sampled during night with the increasing temperature during the day. However, during morning-to-morning sampling strategy, the filters will collect the  $NO_3^-$  quantitatively at night, and the higher temperature in the afternoon of the previous day may promote the loss of  $NO_3^-$  from the filter (Malaguti et al., 2015)."

11. 341-342 Why Xact online instrument gave 1.9 times higher than IC?

Reply: Interestingly, IC measurements of Cl- were found to be higher than Xact 625i measurements during summer at IITD and winter at IITMD. The Cl measurements from Xact 625i were ~1.9 times higher than the measurements from IC during winter at IITD. It could be due to the differences in water-soluble fraction of chloride in the samples, as ionic concentration (IC) represents water-soluble fraction whereas elemental concentration (Xact 625i) represents total concentration. Also, a lot particulate bound chloride in the atmosphere is in the form of ammonium chloride (Manchanda et al., 2021). Part of the ammonium chloride collected during the day long offline sampling would have vaporized, giving lower concentration from IC measurements. Further investigation is needed to draw a firm conclusion. We added this in line no 370-381 in MS as-

"We also compared Cl- measurements from Xact 625i with the measurements from IC. Interestingly, IC measurements of Cl- were found to be higher than Xact 625i measurements during summer at IITD and winter at IITMD. The Cl measurements from Xact 625i were ~1.9 times higher than the measurements from IC during winter at IITD (see fig. SM2 in supplementary material). The correlations were found to be good during winter  $(R^2 = 0.83 \text{ and } 0.76 \text{ at IITD} \text{ and IITMD} \text{ respectively})$  and worsen during summer  $(R^2 = 0.27 \text{ at IITD})$ , similar to what we observed for AMS\_Cl- and IC\_Cl-. It could be due to the differences in water-soluble fraction of chloride in the samples, as ionic concentration (IC) represents water-soluble fraction whereas elemental concentration (Xact 625i) represents total concentration. Also, a lot particulate bound chloride in the atmosphere is in the form of ammonium chloride (Manchanda et al., 2021). Part of the ammonium chloride collected during the day long offline sampling would have vaporized, giving lower concentration from IC measurements. Further investigation is needed to draw a firm conclusion."

12. 346 What were the cut-offs of online instruments?

Reply: The cut-offs of both HR-ToF-AMS and Xact 625 was 2.5µm.

13. Fig 4. Online and offline instrument results could be better marked to separate them. You can't see well the smallest concentration.

Reply: We divided the elements in the plots, based on their concentration ranges. We agree that the smallest concentration can't be separated well compared to its adjacent concentration. We reorganised some elements but that could not work out as well. For example the difference in concentration of online V vs offline V and online Ni vs offline Ni will still be present if we make a separate plot for them (see Fig. SM4 (d)). In Fig. 4, we only kept some major elements and the box plots for rest of the elements were placed in the supplementary material as suggested by referee #3. We modified Fig. 4. In the MS as-

Fig.4. Box plots of some major elements measured offline and online during (a) summer campaign at IITD, (b) winter campaign at IITD, and (c) winter campaign at IITMD site. The box plots for rest of the heavy and trace elements are shown in fig. SM4 of the supplementary material.

14. Fig 5. The highest shares are easy to separate, but the smallest cannot be seen.

Reply: We agree with the reviewer that the smallest shares cannot be seen in the pies. In the modified plots the lowest shares are grouped and named as 'others' beside the highest shares. The elements, that consisted 'others' and their shares are also annotated.

---

## Author Comment (AC2)

**Inter-comparison of online and offline methods for measuring ambient heavy and trace elements and water-soluble inorganic ions (NO$_3^-$, SO$_4^{2-}$, NH$_4^+$ and Cl$^-$) in PM$_{2.5}$ over a heavily polluted megacity, Delhi**

Himadri Sekhar Bhowmik[1], Ashutosh Shukla[1], Vipul Lalchandani[1], Jay Dave[2], Neeraj Rastogi[2], Mayank Kumar[3], Vikram Singh[4], and Sachchida Nand Tripathi[1*]

[1] Department of Civil Engineering, Indian Institute of Technology Kanpur, Kanpur, India

[1*] Department of Civil Engineering and Centre for Environmental Science and Engineering, Indian Institute of Technology Kanpur, Kanpur, India

[2] Geosciences Division, Physical Research Laboratory, Ahmedabad, India

[3]Department of Mechanical Engineering, Indian Institute of Technology Delhi, New Delhi, India

[4]Department of Chemical Engineering, Indian Institute of Technology Delhi, New Delhi, India

*Correspondence to:* S. N. Tripathi (snt@iitk.ac.in)

**Responses (text in blue) to comments by the reviewer (text in black)**

We thank the referee for his/her valuable comments which have greatly helped us to improve the manuscript. Please find below our point-by-point responses (in blue) after the referee comments (in black). The changes or existing lines in the revised manuscript are written in *italic (red)*.

**Referee #3**

**General Comments:** Bhowmik et al. presents an inter-comparison for on- and off-line measurements of water-soluble inorganic ions and heavy and trace metals during two different periods in the Delhi NCR region of India. The on-line measurements included an Aerodyne High-Resolution Aerosol Mass Spectrometer (herein AMS) for water soluble inorganic ions and the Xact 625i Ambient Metals Monitor for heavy and trace metals. The off-line measurements involved collecting aerosols on quartz filters, prepared (depending on what was being extracted), and analyzed by ion chromatography (IC) for water soluble inorganic ions or inductively coupled plasma mass spectroscopy (ICP-MS) for heavy and trace metals. Though of potential interest to the AMT community, especially as the study covers different seasons and a polluted megacity, there are many concerns that need to be addressed prior to publication, as discussed below and outlined by the other reviewer.

1) As this is a techniques paper and is comparing different methods to measure aerosol, more information needs to be added concerning the instruments. For example, the authors say that more details about the AMS can be found in Lalchandani et al. (2021, under review). As it is under review, it is difficult to understand how the AMS was ran and analyzed and would be beneficial for this manuscript to be included, at minimum briefly. This includes:

Reply: We thank the referee for the suggestions. At the time of submitting, the referred research paper was under review but Lalchandani et al. (2022) has been published now. We addressed the referred queries and added to the MS in Line no 230-237 as-

*"A recommended collection efficiency (CE) of 1 (Hu et al., 2017) was used for capture vaporizer. At the beginning and mid of each campaign at the two sites, ionization efficiency (IE) calibration was performed by injecting mono-disperse 300 nm ammonium nitrate and ammonium sulphate particles into AMS and a condensation particle counter (Jayne et al., 2000). A relative ionization efficiency (RIE) of 4.05 and 4.35 was used for IITD and IITMD site, respectively in case of NH$_4$. RIE of SO$_4$ was taken as 2.89 and 1.67 for IITD and IITMD, respectively. For Org and Cl, by default a RIE of 1.4 and 1.3 were taken, respectively. More details can be found in (Lalchandani et al., 2022)."*

1a) How frequently was IE conducted? How stable was it?

Reply: Ionization efficiency (IE) calibrations for nitrate and sulfate were performed once at the beginning and another in the middle of the campaign, following the CPC mass-based method using ammonium nitrate and ammonium sulfate, respectively. It was quite stable (within 10%) as the IE did not differ too much.

1b) What NH$_4$ RIE was used?

Reply: The relative ionization efficiency (RIE) of $NH_4$ was experimentally calculated to be 4.05 and 4.35 for IITD and IITMD, respectively.

1c) What $SO_4$ RIE was used?

Reply: The relative ionization efficiency (RIE) of $SO_4$ determined from the IE calibrations was taken as 2.89 and 1.67 for IITD and IITMD, respectively.

1d) Was the vaporizer a capture or standard vaporizer? What CE was used?

Reply: The vaporizer was a capture vaporizer.

A collection efficiency (CE) value of 1 was taken following the recommendation of Hu et al. (2017) in the case of capture vaporizer.

2) Another important aspect in better understanding the comparisons includes how the aerosols were sampled. This includes:

2a) How long was the sampling line for each instrument? What was the residence time for each sampling line? What material is used throughout? E.g., line 162-163, it appears that a combination of black silicon tubing and stainless-steel was used.

Reply: For HR-ToF-AMS, at IITD the inlet length was 2.44 meter and it was made of stainless steel having inner diameter of 0.3 inch and outer diameter of 0.4 inch whereas, at IITMD, a 1.5 m long silicon tubing (TSI Inc) having 0.19 inch inner diameter was used as inlet. For Xact 625, at both the sites, the inlet length was 2.44 meter and it was made of aluminium having inner diameter of 1.25 inch. It was mentioned in the line no 169-173 in the MS as-

*"Ambient fine particulate matters were sampled through $PM_{2.5}$ cyclone (BGI, Mesa Labs. Inc.) inlet at IITD with a flow rate of 5 lpm (l/min) using a 2.44 m long stainless-steel tubing (0.3 inch I.D and 0.4 inch O.D) and through black silicon tubing (0.19 inch I.D) at IITMD, placed 1.5 m above the rooftop. A Nafion dryer (MD-110-144P-4: Perma Pure, Halma, UK) was used to dry the ambient aerosols to maintain the output RH at 20%."*

And in line no 178-180 in the MS as-

*"A separate sampling line of 2.44 meter (1.25 inch I.D) for the Xact which was made of aluminium was installed. A heater was set up at the end of the sampling line to ensure 45% RH set point."*

The residence times for sampling line at IITD and IITMD for AMS were 14.33 sec and 20.7 sec, respectively. The residence time for sampling line at both IITD and IITMD for Xact was 6.93 sec.

2b) How close were the inlets for the instruments (same line and split, inlets co-located, etc.)?

Reply: At IITD, ambient aerosols were sampled continuously at a flow rate of 5.0 L per min (lpm) through a $PM_{2.5}$ cyclone (BGI, Mesa Labs. Inc) inlet with stainless steel tubing installed on the rooftop. The steel tubing was connected parallel to a Nafion Dryer (MD-110-144P-4; Perma Pure, Halma, UK) and the combined $NO_x$ (ECOTECH Model: Serinus 40 Oxides of Nitrogen Analyzer) and CO (ECOTECH Serinus 30 CO Analyzer) analyzer (flow rate of 1.61 lpm). Then the Nafion dryer output flow was split between AMS (0.08 lpm) and combined instruments which included an Aethalometer (3.0 lpm) and SMPS (0.3 lpm). The sampling line for the Xact metal monitor was separate but co-located and a heater was set up at the end of the long sampling tube.

At IITMD, a black silicon tubing was used as inlet for AMS and a co-located but separate line was installed for Xact.

As CO and $NO_x$ data were not included in this study we did not add this in MS to avoid confusion.

2c) Were the instruments located in a temperature controlled area? What was the temperature difference between inside and outside, as this could potentially lead to biases in the aerosol (e.g., evaporation or high water content)?

Reply: At both the sites, measurements were conducted inside an air-conditioned laboratory. Before entering the instruments (AMS) , particles were passed through a Nafion dryer (MD-110-144P-4: Perma Pure, Halma, UK) to reduce the relative humidity (RH) to ~20%. The sampling line for the Xact metal monitor was separate and a heater was set up at the end of the long sampling tube. The heater power was adjusted to ensure the 45% RH set point (Xact instrument has temperature and RH sensors) and avoid water deposition at Teflon tape. During summer and winter, the daily average ambient temperature was 33.2±4.6 ℃ and 20.1±6.2 ℃, respectively whereas, the inside temperature was maintained at 25ºC.

This was mentioned in the lines 134-136 in the MS as-

*"Further, one HR-ToF-AMS and Xact ambient metal mass monitor were deployed inside a temperature-controlled laboratory on the 3rd floor of the same building about ~10 m above the ground level."*

And in the lines 143-145 in the MS as-

*"Moreover, HR-ToF-AMS and Xact ambient metal mass monitor were installed inside a temperature-controlled laboratory on the 2nd floor of the same building at the height of ~8 m from the ground level."*

Moreover, we modified the lines 166-173 in the MS as-

*"At both IITD and IITMD site, high-resolution time-of-flight aerosol mass spectrometers (HR-ToF-AMS, Aerodyne Research Inc., Billerica, MA) (Canagaratna et al., 2007; DeCarlo et al., 2006), equipped with $PM_{2.5}$ aerodynamic lens (Peck et al., 2016) (Aerodyne Research Inc., Billerica, MA, USA) were installed inside air-conditioned laboratories. Ambient fine particulate matters were sampled through $PM_{2.5}$ cyclone (BGI, Mesa Labs. Inc.) inlet at IITD with a flow rate of 5 lpm (l/min) using a 2.44 m long stainless-steel tubing (12 mm O.D) and through black silicon tubing (0.19 inch O.D) at IITMD, placed 1.5 m above the rooftop. A Nafion dryer (MD-110-144P-4: Perma Pure, Halma, UK) was used to dry the ambient aerosols to maintain the output RH at 20%."*

And in lines 178-182 in the MS as-

*"Two Xact 625i ambient metal monitors (Cooper Environmental Services, Beaverton, Oregon, USA) were installed inside temperature controlled laboratories at IITD and IITMD. Ambient aerosols were sampled through a PM2.5 inlet with a flow rate of 16.7 lpm. A separate sampling line for the Xact was made and a heater was set up at the end of the sampling line to ensure 45% RH set point."*

2d) Was a denuder used for the offline sampling?

Reply: No, during the offline sampling denuder was not being used.

2e) Line 161, the authors mentioned they operated a $PM_{2.5}$ cyclone in front of the $PM_{2.5}$ lens AMS. Did they operate the $PM_{2.5}$ cyclone at different flow rates to ensure the cyclone had a $d_{50} > PM_{2.5}$ or did they operate the cyclone normally? If the latter (normally), the aerosol being measured by the AMS will be much less than $PM_{2.5}$ as the combination of the $PM_{2.5}$ cyclone (operated normally) and $PM_{2.5}$ lens would significantly cut-off the large particles.

Reply: Ambient aerosols were sampled continuously at a flow rate of 5.0 L per min (lpm) through a $PM_{2.5}$ cyclone (BGI, Mesa Labs. Inc) inlet with stainless steel tubing (0.8 cm ID and 1 cm OD) installed on the rooftop. The cyclone from BGI MESA Labs has a PM2.5 cutoff at 16.7 lpm. We operated it at 5 lpm to make the cyclone cutoff go to much larger particle sizes similar to URG cyclone. The URG cyclone (http://www.urgcorp.com/products/inlets/anodized-aluminum-cyclones/urg-2000-30egn-a) indicates that lowering flow rates allow to go to cutoffs of 4.5 micron at flow rates of 10 lpm. The same trend was followed by the BGI MESA cyclone. Thus we operated the flow rate at 5 lpm to make sure the cyclone had a $d_{50} > PM_{2.5}$. The particles were then sampled at a flow rate of 0.08 litres per minute (lpm) through a 100 µm critical orifice and focused into a narrow beam through an aerodynamic lens, to sample $PM_{2.5}$ particles with a transmission efficiency of >50% (Peck et al., 2016). The experiment setup was similar to Tobler et al. (2020), Shukla et al. (2021) and Lalchandani et al. (2022). The details can be found in our companion paper Lalchandani et al. (2022).

2f) Was a cylcone or impactor used to collect the aerosol on the filters? Was there a dryer in front of the filters?

Reply: Filter samples were collected using a High Volume Air Samplers (TISCH environmental) in which the operation is based on impaction.

No, there was no dryer attached in front of the filters.

3) There is some potential concern in the AMS interpretation. Within the AMS community, it is well established that the $NH_4$, $SO_4$, and $NO_3$ signal is a combination of organic and inorganic aerosol (e.g., Farmer et al., 2010, Almeida et al., 2013, Fry et al., 2013, Ge et al., 2014, Kiendler-Scharr et al., 2016, Chen et al., 2019, Schueneman et al., 2021, Nault et al., 2021, Day et al., 2022). Thus, there could be some nuances that are not considered in the direct comparison of the inoragnic aerosol observed by IC vs the $NH_4$, $SO_4$, and $NO_3$ observed by AMS, especially during biomass burning events (e.g., nitrocatechol, Finewax et al., 2018). Further, as the AMS has difficulty

observing refractory chloride (e.g., Tobler et al., 2020), is there a way to filter the observations of chloride to better compare? However, as a different on-line measurement showed even higher chloride measurements, it is currently unclear all the sources of chloride here (e.g., line 341).

Reply: We thank the referee for the comment. We agree that the inorganics data from the AMS could have some interferences from organics but $NO_3$, $SO_4$ and, $NH_4$, were quantified using high mass resolution (HR peak fitting), thus the interferences from organic should be minimum. This is widely accepted. We added the lines 318-322 in the MS as-

*"Some studies suggested that the $NH_4$, $SO_4$ and, $NO_3$ measured by AMS have interferences from organics and can often be misinterpreted as fully inorganic (Chen et al., 2019; Farmer et al., 2010; Day et al., 2021). Though the interference is minimum, this could lead to possible biases between the online and offline measurement of the inorganics."*

AMS only measures the NR-Cl$^-$ and has difficulty observing refractory-Cl$^-$. We tried to find out the possible reasons of this bias by comparing Cl$^-$ from IC with measurements from another online instrument, Xact 625i, which measures Cl using XRF technique. IC measurements of Cl$^-$ were found to be higher than Xact 625i measurements during summer at IITD and winter at IITMD. Interestingly, the Cl measurements from Xact 625i were ~1.9 times higher than the measurements from IC during winter at IITD. It could be due to the differences in water-soluble fraction of chloride in the samples, as ionic concentration (IC) represents water-soluble fraction whereas elemental concentration (Xact 625i) represents total concentration. Also, a lot particulate bound chloride in the atmosphere is in the form of ammonium chloride (Manchanda et al., 2021). Part of the ammonium chloride collected during the day long offline sampling would have vaporized, giving lower concentration from IC measurements. Further investigation is needed to draw a firm conclusion.

4) Though it is important to document the biases in the aerosol collected on filters, it is not clear what is novel in this analysis concerning the water soluble inorganic ions. Artifacts from filter measurements have been well documented for decades (Klockow et al., 1979, Hayes et al., 1980, Koutrakis et al., 1988, Hering and Cass, 1999, Chow et al., 2005, Nie et al., 2010, Liu et al., 2014, 2015, Heim et al, 2020, Nault et al., 2020). Please be more intentional in explaining the novelty of this analysis.

Reply: We agree with the referee that artifacts from filter measurements have been well documented, but the degree of artifacts can be affected by several factors, including temperature, relative humidity, type of filter substrate, the aerosol loading on the filter substrate, etc. Biases are important to study, as biases can be different in different scenario and biases from artifacts can impact the measured chemical composition. The published studies have compared inorganics mostly from semi-continuous methods with filter-based measurements which are limited to few months of data and fewer sites. In this study, the spatio-temporal measurement of inorganics were captured by a high-resolution AMS and compared with the filter-based measurements from ion chromatography in an enormously polluted area- Delhi NCR. In detail, this study demonstrates a comparison between online and offline measurements of WSIS and heavy and trace metals at two sites in Delhi NCR during summer (June-July 2019), characterized by moderate levels of local pollution and winter (October-December 2019), affected by high levels of pollution from local sources and regional transport of massive crop residue burning emissions from adjoining state of Haryana and Punjab.

5) The discussion about filter sulfate being higher than AMS sulfate is confusing in regards to the reaction of $NH_3$ with $H_2SO_4$ to form $(NH_4)_2SO_4$. Various studies have shown that sulfate can be measured as sulfuric acid from filters and compare well with on-line measurements (Klockow et al., 1979, Hayes et al., 1980, Koutrakis et al., 1988, Nault et al., 2020). Instead, this reaction leads to the off-line measurement being biased high in $NH_4$ compared to on-line measurments (Nault et al., 2020). Rather, could it mainly be due to refractory sulfate (e.g., potassium or calcium sulfate) or as the authors pointed out chemistry occurring on the filters (line 295, $SO_2$ reacting with alkaline particles)?

Reply: We agree with the referee that the reason behind filter sulfate being higher than AMS sulphate could mainly be due to refractory sulfate (e.g., potassium or calcium sulfate). But we believe that the higher sulfate concentrations on the un-denuded offline filter-based measurements were possibly because of the positive sampling artifact. The $SO_2$ is absorbed on the filter by the collected alkaline particles. The higher concentration could also be due to the formation of ammonium bi-sulfate or ammonium sulfate because of the reaction between gas-phase ammonia with the acidic aerosols. Another possible reason for higher filter sulphate could be un-denuded filter measurements. The interaction of ammonia with acidic aerosols can be minimized by using denuders while collecting aerosols on the filters (Nault et al., 2020). This has been added in the MS in line 311-318 as-

*"The higher $SO_4^{2-}$ concentrations on the un-denuded offline filter-based measurements could be due to refractory sulfate (e.g., potassium or calcium sulfate). The higher filter-sulfate could also be because of the positive sampling artifact. The $SO_2$ is absorbed on the filter by the collected alkaline particles (Nie et al., 2010). The higher concentration could also be due to the formation of ammonium bi-sulfate or ammonium sulfate because of the reaction between gas-phase ammonia with the acidic aerosols (Nicolás et al., 2009). Also, the un-denuded filter measurements could lead to higher filter-sulfate. The interaction of gas phase ammonia with acidic aerosols can be minimized by using denuders while collecting aerosols on the filters (Nault et al., 2020)."*

6) The overall general concern about the paper is that the authors present the comparisons; however, they do not really present either steps forward to improve the comparisons or which method may be improved.

Reply: We agree with the referee that the steps forward to improve the comparisons or which method may be improved should also be presented. This has been briefly added in the MS in lines 544-557 as-

*"The above findings highlight the measurement methods' accuracy and implement the particular type of measurements as needed. Denuders could be effective in avoiding the overestimation problems of ammonium and sulphate in filter measurements and improve the comparison. Also, teflon filters instead of quartz filters in the un-denuded sampler are reported to give better comparison for sulphate. The MDLs in the Xact 625 measurements are higher than the MDLs for the offline method. Depending on the objective of the campaign, Xact 625 can be deployed for a longer time interval to analyse the elements that are below their MDLs. The high resolution real-time monitoring of non-refractory organics, inorganics by HR-ToF-AMS and elements by Xact comes at the cost of high sensitivity in MDLs, calibrations and cost. Whereas, cost effectiveness of conventional samplers makes it practical to deploy in larger numbers at multi-sites simultaneously. Overall, high resolution real time sampling provides a rich dataset for high and small pollution episodes. Future work should involve using different filter substrates and different digestion protocols to re-evaluate the difference between these online and offline methods. Although this study compares the PM species, a comparison of full source apportionment analysis between online and offline methods should be done for more qualitative and quantitative insights."*

7) Another concern is that the authors discarded data that was below 3xMDL. This can artificially raise the average value of the observations (potentially leading to the on-line measurement 24 hr avg being higher than the filter measurement). There is inherent noise in measurements that can be above and below MDL and should not be discarded for that. At minimum, the authors should investigate whether removing this data leads to differences in the comparisons or not.

Reply: We did not discarded all the data below 3 x MDL for all the elements. We rephrased the sentence in line 419-423 in the MS as-

*"Elements having data below $3 \times MDL$ were discarded from further examination and un-reliable as values below $3 \times MDL$ would lead to higher uncertainty (Furger et al., 2017). The elements K, Ca, Ti, Mn, Fe, Ba, and Pb have >80% of their values above both offline and online MDLs, and thus the data quality is reliable. Further, Ni, Mo, Zr have higher blank concentrations, and thus the data is not reliable for ICP-MS measurements."*

We investigated as per the referee suggested and found out that the slope and $R^2$ vary within ~10% for the elements like Al, K, Ca, Ti, Mn, Fe etc which have >80% values above both method's MDLs. For example, the slope (online/offline) and $R^2$ for Al during summer at IITD was 0.25 and 0.72 while the data below 3 x MDL were not removed whereas, the slope and $R^2$ for Al increased to 0.31 and 0.81 while the data was removed.

8) I agree with Reviewer #1 that the discussion about when the filter is sampled/collected, as written, is very confusing and makes it unclear what is the best method. Also, did the authors try both methods to verify this?

Reply: We have re-written the part in a logical way which is now easier to follow in line 329-346 in MS as-

*"The online and offline $NO_3^-$ measurements posed a good correlation during winter ($R^2 = 0.91$ and slope of 1.07 at IITMD, $R^2 = 0.82$ and slope of 0.49 at IITD) whereas the correlation worsens during summer at IITD ($R^2 = 0.42$ and slope of 1.78) (Fig. 3). The slopes and correlation coefficient for the WSIS are listed in Table 2. The $NO_3^-$ concentrations measured by the HR-ToF-AMS were higher than the offline data during summer at IITD and during winter at IITMD whereas, filter-based measurements of $NO_3^-$ were higher during winter at IITD (Fig. 2). The higher offline $NO_3^-$ concentrations during winter at IITD can be possibly because of the positive artifact due to the absorption of gas-phase nitric acid ($HNO_3$) on the filter (Chow, 1995). Many studies (Chow et al., 2008; Kuokkaet al., 2007; Malaguti et al., 2015) reported higher concentrations of $NO_3^-$ from high time resolution measurements than filter-based measurements due to the evaporation of ammonium nitrate collected on filters over the duration of sample collection (Pakkanen & Hillamo, 2002; Schaap et al., 2004; Kuokkaet al., 2007). Pandolfi et al., (2014) observed $NO_3^-$ _ HR-AMS/Filter ratios of ~1.7 at Barcelona and Montseny in Europe. This*

*evaporation loss increases with the decrease of humidity and the increase of temperature (Chow et al., 2008; Takahama et al., 2004). Also, complete evaporation may occur beyond 25°C (Schaap et al., 2004).Chow et al., (2008) observed the evaporation loss from quartz filter to be more than 80% during the warm season in central California. The high temperature (35°C-48°C) during the long sampling hours (24 hours) may be a possible reason for the poor correlation between online and offline NO$_3^-$ measurements during the summer campaign at IITD."*

We also have re-written the part in line 346-352 in MS as-

*"Schaap et al. (2004) reported that the NO3- volatilization during a 24-h sampling period not only depends on the sampling instruments and ambient conditions, but also on sampling strategy. If the sampling strategy is evening to evening (24 hours), the samples will lose the NO3- sampled during night with the increasing temperature during the day. However, during morning-to-morning sampling strategy, the filters will collect the NO3- quantitatively at night, and the higher temperature in the afternoon of the previous day may promote the loss of NO3- from the filter (Malaguti et al., 2015)."*

9) Many of the figures are too busy and difficult to interpret. E.g., Fig. 6a-c could go into the SI and keep Fig. 6d in the main document, as this best summarizes the results. Further, for Fig. 3 and 6d, the authors may consider adding on the right axis what the R$^2$ is for each slope. Further, it is unclear what Fig. 5 adds to the discussions in the paper. Finally, for Fig. 4, the authors may consider only showing one or two of the important subplot and placing the rest into the SI.

Reply: We partly agreed with the referee. We do, however, not agree with removing Fig. 6a-c from MS and keeping in the supplementary material, as it demonstrates the categorised element's groups according to their comparable characteristics (A, B, and C) at three locations in two different seasons which cannot be demonstrated by Fig. 6d alone. We also modified the Fig.6 as per the suggestion given by referee #1.

We modified Fig. 3e and 6d as per the given suggestion and changed in the MS as-

[Figure]

*Fig.3. ……(e) comparison of slopes (online/offline) and R$^2$ of the measured inorganic ions in PM$_{2.5}$ during summer and winter campaign at IITD and during winter campaign at IIITMD.*

[Figure]

*Fig. 6. …… (d) comparison of slopes (online/offline) & R² of the measured heavy and trace metals in PM₂.₅ during the summer and winter campaign at IITD and winter campaign at IIITMD.*

Where Fig. 4. shows the comparison of the elements absolute value, the comparison of fractions of the elements in total element concentration for both the measurements during summer and winter campaign at IITD and during winter campaign at IITMD can be find in Fig. 5. We consider Fig. 5 relevant for the article.

In Fig. 4, we only kept some major elements and the box plots for rest of the elements were placed in the supplementary material as suggested. We modified Fig. 4. In the MS as-

[Figure]

*Fig.4. Box plots of some major elements measured offline and online during (a) summer campaign at IITD, (bd) winter campaign at IITD, and (c) winter campaign at IITMD site. The box plots for rest of the heavy and trace elements are shown in fig. SM4 of the supplementary material.*

10) A table that summarizes the measurements, their LOD and uncertainty, and their size cut off would be beneficial.

Reply: The cut-off of both HR-ToF-AMS and Xact 625 was 2.5µm. The measurements, standard deviations (uncertainty) and their MDLs were shown in Table SM1 and SM2 in supplementary material.

11) Please read through the manuscript again. There are some grammatical concerns and things that need to be defined to improve readability. Some examples are included below:

11a) Line 27-28, IITD and IITMD are not known and need to be defined.

Reply: The sentence has been rephrased in line 27-29 in MS as-

*"The comparison was performed over two seasons (summer and winter) and at two sites (Indian Institute of Technology, Delhi and Indian Institute of Tropical Meteorology, Delhi) which are located in Delhi, NCR, India, one of the heavily polluted urban areas in the world."*

11b) Line 133, cum?

Reply: Rephrased as *"as well as"*.

11c) Line 163, capitalize r in Research

Reply: We thank the reviewer for pointing out this mistake. The same has been corrected in the revised manuscript.

11d) Line 228, 6N?

Reply: 6N referred to 6 Normal $HNO_3$, which defines the concentration of $HNO_3$ used.

11e) Fig. 2, symbols, box not defined as in Fig. 4

Reply: Plots modified with defined box as-

[revised manuscript text omitted]

---

## Author Response (AR2)

**Inter-comparison of online and offline methods for measuring ambient heavy and trace elements and water-soluble inorganic ions ($NO_3^-$, $SO_4^{2-}$, $NH_4^+$ and $Cl^-$) in $PM_{2.5}$ over a heavily polluted megacity, Delhi**

Himadri Sekhar Bhowmik[1], Ashutosh Shukla[1], Vipul Lalchandani[1], Jay Dave[2], Neeraj Rastogi[2], Mayank Kumar[3], Vikram Singh[4], and Sachchida Nand Tripathi[1*]

[1] Department of Civil Engineering, Indian Institute of Technology Kanpur, Kanpur, India

[1*] Department of Civil Engineering and Centre for Environmental Science and Engineering, Indian Institute of Technology Kanpur, Kanpur, India

[2] Geosciences Division, Physical Research Laboratory, Ahmedabad, India

[3] Department of Mechanical Engineering, Indian Institute of Technology Delhi, New Delhi, India

[4] Department of Chemical Engineering, Indian Institute of Technology Delhi, New Delhi, India

*Correspondence to:* S. N. Tripathi (snt@iitk.ac.in)

**Responses (text in blue) to comments by the reviewer (text in black)**

We thank the referee for his/her valuable comments which have greatly helped us to improve the manuscript. Please find below our point-by-point responses (in blue) after the referee comments (in black). The changes or existing lines in the revised manuscript are written in *italic (red)*.

**Referee #3**

**General Comments:** Bhowmik et al. present comparisons between on- and off-line measurements of both refractory and non-refractory aerosol. They show differences between the measurements, related to either the limitations of the instrument (e.g., AMS only observing non-refractory aerosol) or known interferences with the technique (e.g., volatilization / reactions).

The authors have taken great care in addressing the majority of the comments from the reviewers. This in turn has generally improved the manuscript and the figures. However, there are still a few details that need to be improved upon for the paper to be accepted to AMT.

**Comment 1:** Page 13, lines 322 - 325: Sorry for miscommunicating this point. The concern is not for the inorganic ions having interference with organic ions (though that is a concern for m/z 30). The concern is in regards that the organic sulfate / nitrate / reduced nitrogen will thermally decompose and / or under electron ionization, produce that inorganic ion (e.g., for organic nitrate, a large fraction of the signal is NO+ and NO2+ (Day et al., 2022) and for organic sulfates, a large fraction of the signal is SO+, SO2+, and SO3+ (Chen et al., 2019)). At minimum, please rephrase those lines to reflect this aspect of the AMS.

Response: We thank the reviewer for communicating this point. The lines has been rephrased in the MS in lines 318-323 as-

*"Some studies suggested that the organic sulfate, nitrate, and reduced nitrogen thermally decompose and/or under electron ionization in AMS, producing inorganic ions. $NO^+$ and $NO_2^+$ contribute a significant fraction in organic nitrate (Day et al., 2022) whereas, for organic sulfates, a large fraction of the signal is contributed by $SO^+$, $SO_2^+$, and $SO_3^+$ (Chen et al., 2019). Though the additional inorganics are minimum, this could lead to possible marginal biases between the online and offline measurement of the inorganics."*

**Comment 2:** Page 16, line 365 - 367: It is still unclear if the filters are so warm from the afternoon that they cause lost of ammonium nitrate collecting during the night.

Response: In the morning-to-morning sampling strategy, the filters changed in the morning will sample nitrate quantitatively throughout the night as evaporation will be less at a relatively lesser temperature during the night. The nitrate collected during the afternoon of the previous day will be lost at the higher temperatures during noon-afternoon (20°C-25°C during the winter campaign and 38°C-45°C during the summer campaign). While the filters changed in the afternoon or evening may lose the nitrate sampled during the night since the evaporation increases with the increase of temperatures during the day. For better understanding, we rephrased the lines in the MS in lines 349-353 as-

*"If the sampling strategy is evening-to-evening (24 hours), the samples will lose the night $NO_3^-$ sampled during the night with the increasing temperature during the day. However, during morning-to-morning sampling strategy, the filters will collect the night $NO_3^-$ quantitatively throughout the night, but the higher temperature in the afternoon of the previous day may promote the loss of afternoon $NO_3^-$ from the filters (Malaguti et al., 2015)."*

**Comment 3:** Minor but of use: I understand the meteorology for the location of the measurement was not working; however, were there other meteorological stations in the city that could be used? It is not critical for the paper or for it to be accepted; instead, it would just improve it.

Response: We understand that the meteorology would have improved the manuscript. We looked through data from the nearby meteorological stations managed by Central Pollution Control Board (CPCB), India but, the quality of the data was poor. Hence we did not add the data to our manuscript.